# Negative reciprocity, not ordered assembly, underlies the interaction of Sox2 and Oct4 on DNA

John W Biddle, Maximilian Nguyen, Jeremy Gunawardena*

Department of Systems Biology, Harvard Medical School, Boston, United States

**Abstract** The mode of interaction of transcription factors (TFs) on eukaryotic genomes remains a matter of debate. Single-molecule data in living cells for the TFs Sox2 and Oct4 were previously interpreted as evidence of ordered assembly on DNA. However, the quantity that was calculated does not determine binding order but, rather, energy expenditure away from thermodynamic equilibrium. Here, we undertake a rigorous biophysical analysis which leads to the concept of reciprocity. The single-molecule data imply that Sox2 and Oct4 exhibit negative reciprocity, with expression of Sox2 increasing Oct4's genomic binding but expression of Oct4 decreasing Sox2's binding. Models show that negative reciprocity can arise either from energy expenditure or from a mixture of positive and negative cooperativity at distinct genomic loci. Both possibilities imply unexpected complexity in how TFs interact on DNA, for which single-molecule methods provide novel detection capabilities.
DOI: https://doi.org/10.7554/eLife.41017.001

## Introduction

The transcription factors (TFs) Sox2 and Oct4 work in concert to specify the earliest lineages in the mammalian embryo (*Pesce and Schöler, 2001*; *Avilion et al., 2003*). Chen et al. previously undertook an extensive single-molecule analysis of the dynamics of Sox2 and Oct4 in living cells to determine how these TFs found their cognate sites on DNA (*Chen et al., 2014*). One of their main conclusions was a preferred order of assembly on DNA, with binding of Sox2 followed by Oct4 in approximately $75\%$ of instances (*Figure 1A*). They justified this claim by determining the ratio,

$$R = \frac{K_1 K_3}{K_2 K_4}, \tag{1}$$

where the values of the $K_i$ were obtained directly from their single-molecule data. Chen et al. interpreted each $K_i$ in terms of the model in *Figure 1A*, as the ratio of the binding rate to the unbinding rate of the corresponding transition (we use a different convention here to that actually followed by Chen et al.; see below). They asserted that $R$ was the ratio of the probability of taking the upper pathway in *Figure 1A*, in which Sox2 binds first, to the probability of taking the lower pathway, in which Oct4 binds first. Their data showed that $R = 3.05$, leading to the binding-order frequency shown in *Figure 1A* and the claim of ordered assembly.

However, the quantity $R$ in *Equation 1* is not the probability ratio of the two binding pathways. This ratio can be calculated, as explained below, but what $R$ determines instead is the 'cycle condition' for the model in *Figure 1A*. If this model is assumed to be at thermodynamic equilibrium, so that no external sources of energy are being consumed, then, as a matter of fundamental physics, $R = 1$. Under the assumptions made by Chen et al., the value $R = 3.05$ immediately implies the presence of energy-expending mechanisms acting 'behind the scenes' that are maintaining the system described by *Figure 1A* away from thermodynamic equilibrium.

*For correspondence:
jeremy_gunawardena@hms.
harvard.edu

Competing interests: The authors declare that no competing interests exist.

**eLife digest** The bodies of humans and other animals contain many types of cells that perform different roles in the body. Most cells in the body carry the same DNA, which is arranged into sections known as genes. The marked differences between cell types arise because different sets of genes are switched on or 'expressed'.

Proteins called transcription factors control which genes are expressed by binding to DNA and recruiting groups of accessory proteins. However, it is not clear how they interact with each other and with the accessory proteins to decide whether to express a gene. For instance, it is thought that some accessory proteins may provide energy for this process, but it is unknown whether the energy is used continuously or only for a short time. Insights from physics suggest that the former could have more powerful effects.

In 2014, a team of researchers reported using a microscopy approach, known as single-molecule imaging, to follow two transcription factors called Sox2 and Oct4 in cells from mice. After analyzing the data, the researchers concluded that Sox2 and Oct4 had a specific order of binding to DNA, with Sox2 often binding first and then assisting Oct4 to bind. Now Biddle et al. report that the claim made in the 2014 study is unsupported because of errors in the way the data were analyzed. In particular, Biddle et al. argue that what the earlier study actually calculated is not an order of binding but a measure of whether energy is being continuously used when Sox2 and Oct4 bind to DNA.

Biddle et al. reanalyzed the data from the 2014 work and concluded that Sox2 increases the extent of Oct4 binding to DNA, while Oct4 decreases the amount of Sox2 binding to DNA. Mathematical models suggest this may be due to the continuous use of energy as the two proteins bind to DNA. Alternatively, it could also happen if Sox2 and Oct4 helped each other to bind at some sites on DNA and hindered each other from binding in other places, even if energy is only used for a short time. These findings reveal that there is unexpected complexity in how transcription factors bind to DNA.

The next step following on from this work is to carry out experiments that test the two possible explanations for how Sox2 and Oct4 interact on DNA. Including physics in the analysis may help describe more accurately the biology of how transcription factors determine gene expression. Understanding this process will shed new light on many important biological questions and may aid the development of gene therapy and other new medical techniques.

DOI: https://doi.org/10.7554/eLife.41017.002

The role of energy expenditure in regulating eukaryotic genes is especially interesting. One of the most striking differences between eukaryotic genomes and eubacterial ones is the presence of multiple energy-expending mechanisms, which reorganise chromatin, remodel nucleosomes and post-translationally modify regulators. Although much is known about the molecular components involved in such energy transduction, the functional significance of energy expenditure has been slow to emerge. There is debate, for instance, as to whether pioneer TFs, like Sox2 and Oct4, can open chromatin without relying on external sources of energy (*Cirillo et al., 2002*; *Iwafuchi-Doi and Zaret, 2016*) or whether they recruit ATP-dependent chromatin remodellers to undertake this (*Voss et al., 2011*; *Swinstead et al., 2016*; *Zaret et al., 2016*).

Physics provides fundamental insight into the significance of energy expenditure. As first pointed out by Hopfield for replication, transcription and translation, if the underlying biochemical system is operating at thermodynamic equilibrium, the cycle condition ($R = 1$) sets a fundamental limit below which the error rate cannot be reduced (*Hopfield, 1974*). This limit is much higher than the very low error rates observed in practice and Hopfield introduced kinetic proofreading as an energy-expending mechanism which could account for this enhanced functionality. More recently, we have identified the analogous 'Hopfield barrier' for sharpness in gene regulation and suggested that energy expenditure may be implicated in the sharp response of early developmental genes (*Estrada et al., 2016*). From a broader viewpoint, following the energy may be a powerful strategy for making functional sense of the molecular complexity underlying eukaryotic gene regulation.

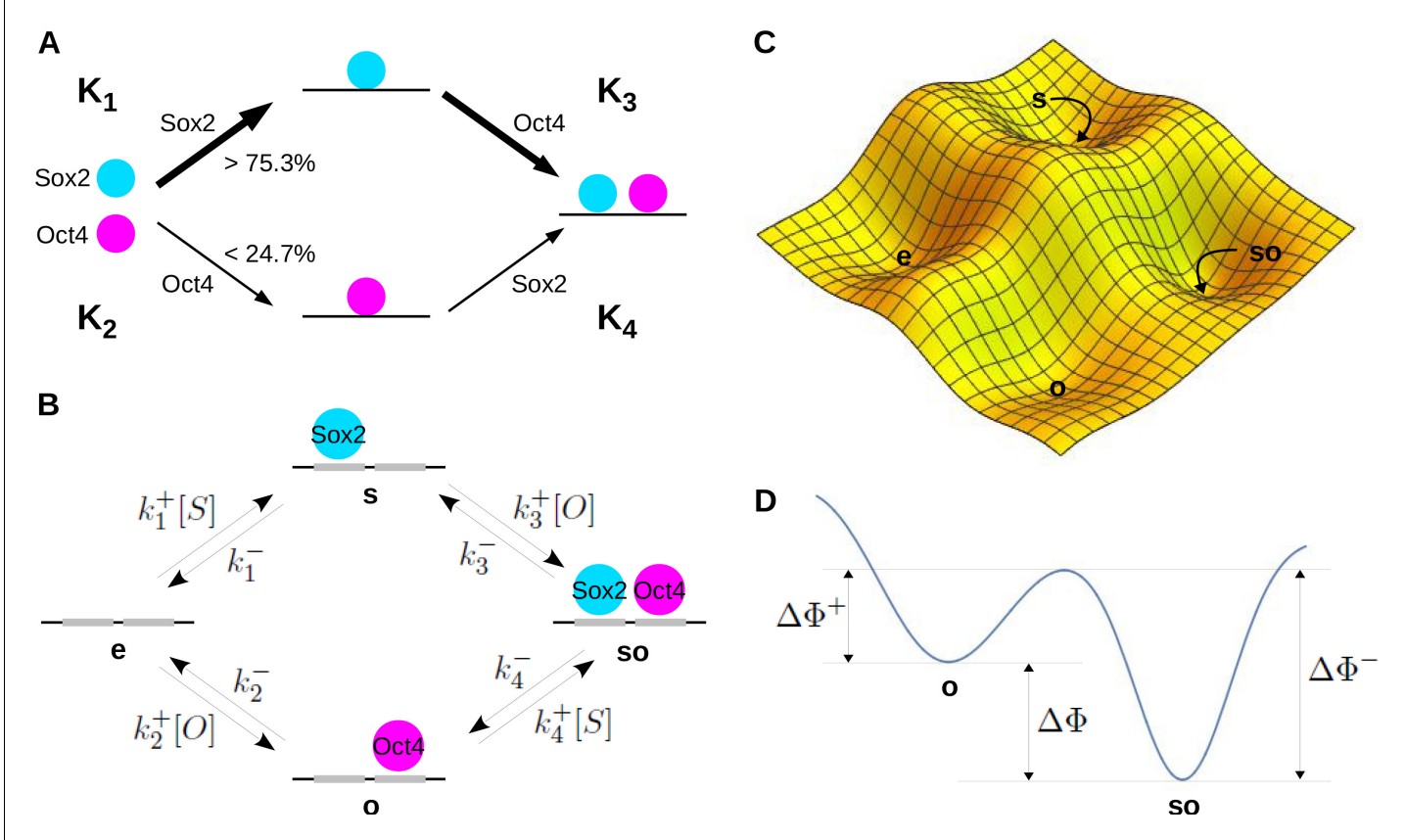

**Figure 1.** Ordered assembly, graphs and energy landscapes. (**A**) Ordered assembly of Sox2 and Oct4 on DNA adapted from (*Chen et al., 2014*, Figure 6E) and annotated with the quantities $K_i$ described in the Introduction. (**B**) Corresponding linear framework graph for Sox2 and Oct4 binding to DNA, showing the microstates (vertices), transitions (directed edges) and rates (edge labels). (**C**) Hypothetical 2D energy landscape for the system in A, showing the free-energy minima corresponding to the four microstates. (**D**) Hypothetical 1D cross-section through the energy landscape, along a reaction coordinate that traverses the lowest point on the 'continental divide' between microstates **o** and **so**, as explained in the text. The annotations show the free-energy differences which influence the kinetic labels for binding of Sox2 to **o** ($\Delta\Phi^+$) and for unbinding of Sox2 from **so** ($\Delta\Phi^-$). The thermodynamic ratio of the binding to unbinding label is determined by the difference in the free-energy minima ($\Delta\Phi$), as given by *Equation 3*.
DOI: https://doi.org/10.7554/eLife.41017.003

Given the significance of energy expenditure, it is important to examine further the calculation of $R$ made by Chen et al. This paper is devoted to revisiting their assumptions and reanalysing their data on a rigorous biophysical foundation. In the remainder of this Introduction, we describe in informal language the path that we took and the novel insights that emerged about the interaction of Sox2 and Oct4 on the genome.

As an initial check, we asked how much significance could be given to the estimate by Chen et al. that $R>1$. Based on a conservative statistical analysis of their data (Materials and methods), the probability that $R \leq 1$ is less than $10^{-9}$, so their estimate is highly significant. As for the data themselves, these were acquired by fluorescently labeling each TF and using a variety of powerful single-molecule techniques to follow individual TFs within the nuclei of living cells. These data suggest that the TF moves back and forth between two distinct states, one in which it is specifically bound to sites on DNA and one in which it is not specifically bound. (The latter state is potentially complicated, involving both diffusion and non-specific binding to DNA.) This conclusion is supported by a detailed biophysical analysis of the measurement process and the movement of TF molecules within the nucleus. We have nothing to say about this analysis of TF behavior and take it as our starting point.

Our concern is with drawing conclusions about what happens on DNA. This implies a fundamental shift of viewpoint, from that of the TF, which is being directly observed, to that of the DNA, which is not. This shift requires two ingredients. The first is a model of what happens on DNA and the second

is a method for converting information about TF behavior into information about DNA behavior. For the first, Chen et al. adopted the model in *Figure 1A*. For the second, they assumed that the rates at which the TF moved from being specifically bound to not being specifically bound, and vice versa, as determined from their single-molecule data, were 'good estimates' for the rates at which the TF unbound from, and bound to, DNA, respectively, (*Chen et al., 2014*, page S7). This is correct for the unbinding rate but, as we will explain in more detail below, it is not correct for the binding rate. What the TF 'sees' when it specifically binds to DNA is not the same as what the DNA 'sees' when it becomes specifically bound. It is necessary to develop a rigorous procedure for converting between the TF viewpoint and the DNA viewpoint. This is the first step in our analysis and it may be broadly useful for other studies.

From now on, it will be important to distinguish between the TF viewpoint and the DNA viewpoint. We will use $\kappa_i$ for the numbers which Chen et al. calculated from the TF viewpoint using their single-molecule data. Here, $i = 1, 2, 3, 4$ is the index of the corresponding binding transition in *Figure 1A*. We will use $\rho$ for the corresponding ratio in *Equation 1*,

$$\rho = \frac{\kappa_1 \kappa_3}{\kappa_2 \kappa_4}. \tag{2}$$

To move from the TF to the DNA viewpoint, Chen et al. assumed that $K_i = \kappa_i$, so that $R = \rho$. As pointed out above, this is not correct. We will determine the correct relationship between $\kappa_i$ and quantities like $K_i$ which arise from the DNA viewpoint. This will allow us to interpret the value $\rho = 3.05$ which Chen et al. calculated in terms of what happens on DNA. We will therefore largely be concerned with $\rho$ and *Equation 2* and will refer to $R$ and *Equation 1* only in the context of the model in *Figure 1A*.

The DNA viewpoint model adopted by Chen et al. involves the interplay between two TFs, Sox2 and Oct4 (*Figure 1A*). However, their single-molecule techniques only permit one TF to be observed at a time. They therefore used two cell lines, in each of which one of the TFs was fluorescently labeled and measured while the other TF was inducibly expressed. The quantities $\kappa_1$, $\kappa_2$, $\kappa_3$ and $\kappa_4$ in *Equation 2* were determined from four independent experiments in these two cell lines. This is an interesting strategy whose data provide unexpected biological insights, as we show below. However, it also raises two further difficulties with the analysis made by Chen et al. First, the biological context is no longer accurately described by the model in *Figure 1A*. For one thing, the concentrations of the TFs, which determine their rates of binding to DNA, will differ between the two cell lines and this is not accommodated by *Figure 1A*. Second, Chen et al. assumed that when the second TF was inducibly expressed its binding sites were saturated. This was an oversimplification.

We undertook a more careful analysis to address these issues. We formulated a 'single-locus' model that is derived from *Figure 1A* and which accommodates the experimental context. (The significance of the model name will become clear below.) This model yields an interpretation of the $\kappa_i$ for which we can ask whether $\rho$ in *Equation 2*, can satisfy $\rho \neq 1$ at thermodynamic equilibrium. The answer is yes. For instance, if there is sufficient cooperativity in binding between Sox2 and Oct4, this could yield $\rho \neq 1$ even at equilibrium. Hence, under the single-locus model, $\rho$ in *Equation 2* does not offer evidence for energy expenditure behind the scenes, as implied for $R$ in *Equation 1* under the assumptions made by Chen et al.

However, the single-locus model reveals something further, which leads us to introduce the concept of 'reciprocity'. This is a quantitative measure of how two TFs influence each other's genomic binding. The data acquired by Chen et al. show that Sox2 and Oct4 exhibit negative reciprocity, which means, in this case, that the fractional binding of Sox2 (ie: the proportion of Sox2-binding sites occupied by Sox2) decreases when Oct4 is induced but the fractional binding of Oct4 increases when Sox2 is induced. The statistical significance of negative reciprocity is not as dramatic as for $\rho > 1$—the probability that the reciprocity is not negative is $2 \times 10^{-2}$—but this remains significant enough to require further analysis (Materials and methods). We find that the single-locus model cannot exhibit negative reciprocity at thermodynamic equilibrium but it can do so if the model is away from equilibrium. Hence, on the basis of this model, but by considering reciprocity rather than $\rho$, the data of Chen et al. do provide evidence for energy expenditure behind the scenes.

But there is a further, more subtle problem. Sox2 and Oct4 bind at many loci in the genome, with widely differing strengths and cooperativities. *Figure 1A* and the single-locus model derived from it represent some kind of 'average' view of the underlying genomic diversity. While it is clear how to

average a collection of numbers associated to a series of genomic loci, it is much less clear how to average a collection of models. The mathematical legitimacy of such procedures has hardly been studied. We therefore considered a more general 'genomic-diversity' model, which accommodates two types of loci at which Sox2 and Oct4 can bind with different strengths and cooperativities. We find that if both kinds of loci exhibit positive cooperativity or both exhibit negative cooperativity, then the reciprocity remains positive. However, negative reciprocity can now arise in two ways. First, if the model is away from thermodynamic equilibrium due to energy expenditure behind the scenes and, second, if the model is at equilibrium but the loci exhibit mixed cooperativities, with one kind of locus being positive and the other kind being negative, so that Sox2 and Oct4 both help and hinder each other in binding across the genome. The available data do not distinguish between these two non-exclusive possibilities, either of which could account for the data of Chen et al.

Evidence of mixed cooperativity across the genome or of energy expenditure has not previously been found in living cells and suggests unexpected complexity 'behind the scenes' in how Sox2 and Oct4 interact on DNA. The single-molecule methods of Chen et al. offer exciting capabilities for unravelling such mechanisms.

Two further considerations are important to the broader context of the work described here. First, models are always necessary to draw conclusions from data and the conclusions drawn are thereby contingent on the assumptions underlying those models. These assumptions are subject to interpretation and judgement. This is unavoidable but not always appreciated and we review the implications further in the Discussion.

Second, in contrast to model assumptions, it is a matter of fundamental physics that $R$ in *Equation 1* does not determine binding order for the model in *Figure 1A*. There are no assumptions or interpretations under which $R$ could do so. Furthermore, what $R$ does determine is a matter of substantial interest, which is whether or not the model in *Figure 1A* is away from thermodynamic equilibrium. That these issues have gone unnoticed suggests that the biophysical foundations of gene regulation are not well understood. Indeed, these foundations conceal many subtleties. The cycle condition at thermodynamic equilibrium, also known as 'detailed balance' or 'microscopic reversibility', cannot be derived from classical thermodynamics, in which the concept of thermodynamic equilibrium was first formulated. It rests, instead, on the time-reversal symmetry of the laws of physics (*Tolman, 1938*; *Mahan, 1975*). As for what can happen away from thermodynamic equilibrium, such as the emergence of life, that remains a largely unsolved problem of physics (*Laughlin et al., 2000*; *Ornes, 2017*). If these subtleties reinforce the importance of 'following the energy', they also make clear how essential the biophysical foundations are to that endeavour. Accordingly, we begin with a gentle overview of the foundations. We then build on this to explain the results described above. We have tried to bring out the logic behind the calculations and to lighten the mathematical details as far as possible, relegating many of these to the Materials and methods. We hope in this way to encourage a more rigorous approach to thinking about gene regulation.

## Results

### Biophysical foundations

We describe here the nature of thermodynamic equilibrium, using the model in *Figure 1A* for simplicity, and explain why it implies that $R = 1$. We also introduce the notation and terminology used to analyse the single-locus and genomic-diversity models in subsequent sections.

#### Convention used

Chen et al. defined the quantity $K_i$ in *Figure 1A*, which they denoted $K_{di}$, as the ratio of the unbinding rate to the binding rate of the corresponding transition (*Chen et al., 2014*, pages S6 and S7). For reasons explained below, it is better to use the opposite convention, with the binding rate in the numerator, as above in the Introduction. This cosmetic change has no impact on the arguments presented here but should be kept in mind when comparing what we say here to the text in *Chen et al. (2014)*.

## Linear framework graphs

To make the underlying physics more explicit, we use the 'linear framework' developed in *Gunawardena, 2012*; *Mirzaev and Gunawardena (2013)* and recently applied to gene regulation (*Ahsendorf et al., 2014*; *Estrada et al., 2016*); for a review, see *Gunawardena (2014b)*. One advantage of the framework is that it is based on similar pictures to that in *Figure 1A*, with a biochemical system being described mathematically by a graph with directed edges that carry labels (hereafter, a 'graph'). The graph corresponding to *Figure 1A* is shown in *Figure 1B*. The latter should be viewed as a more mathematically rigorous version of the former.

The vertices of the graph represent the microstates of the system, corresponding, as it were, to snapshots taken from a nanoscopic drone hovering over DNA. The level of resolution, or granularity, can vary; here, only two distinct binding sites on DNA, one for each of the two TFs, Sox2 and Oct4, are shown, with all other details being suppressed. There are 4 microstates, **e**, **s**, **o** and **so**, corresponding to whether or not each of the TFs is bound at its cognate site.

The directed edges of the graph represent stochastic transitions between microstates, as would be seen in a video from our nanoscopic drone. The labels on the edges represent the rates of these transitions with dimensions of (time)$^{-1}$. However, an edge label may be a complicated algebraic expression made up from other terms to reflect how the graph interacts with its environment. We see this in the difference between a label for TF binding, such as $k_1^+[S]$, and a label for TF unbinding, such as $k_1^-$. The former is composed of the product of an 'on-rate', $k_1^+$, and a concentration, $[S]$, while the latter is just an 'off-rate'. On-rates, $k^+$, have dimensions of (concentration $\times$ time)$^{-1}$. In contrast, 'off-rates', $k^-$, are pure rates with dimensions of (time)$^{-1}$.

It is helpful to keep in mind the distinction between binding edge labels, which are pure rates, and the corresponding on-rates, which must be multiplied by a concentration to yield a pure rate. The data of Chen et al. give access to binding labels but not to on-rates because the concentrations of the TFs are not known.

As with all mathematical models, *Figure 1B* conceals many assumptions, of which two appear especially pertinent here. Chen et al. made these same assumptions implicitly. First, we have assumed that the free concentrations of the TFs are not affected by binding or unbinding. If they were, we would have to use molecular numbers rather than concentrations to keep track of the changes. The use of concentrations is a reasonable approximation if there are many more TF molecules than binding sites, which appears to be the case here (*Chen et al., 2014*).

Second, *Figure 1B* shows only a tiny amount of the actual molecular detail that we know to be present, which includes chromatin, co-regulators, histone post-translational modifications and many other features, some of which may be playing a role behind the scenes (as, indeed, our calculations suggest). The models used here are highly 'coarse-grained', with 'effective' parameters that are intended to summarise those molecular complexities which have not been explicitly included. This coarse-graining could conceal properties that only become visible in a more detailed model. Accordingly, the conclusions drawn depend on the model being used. This contingency remains a fundamental aspect of employing models in biology, a point to which we return in the Discussion.

## Steady states of a Markov process

The specification in *Figure 1B* describes what is called a Markov process, which is the mathematical entity that is conventionally used to analyse biochemical systems at the level of single molecules. A Markov process yields a differential equation which describes how the probabilities for being in each microstate change over time. We will not have to use this equation because we will make the assumption that our system is in steady state, so that the microstate probabilities do not change over time. If we return to our drone video footage and repeatedly count the frequencies with which each microstate appears, then the system is in steady state if these frequencies wobble stochastically around average values that do not change over time.

The steady-state assumption is used implicitly throughout the analysis made by Chen et al. It amounts to making a timescale separation in which the system under study is taken to be operating sufficiently fast relative to its environment that it can be assumed to have reached a steady state (*Gunawardena, 2012*). This is always an approximation in biology, but it is a convenient and widely used one because steady states can be analysed far more simply than behaviours that change over time. In the experimental context of Chen et al. and over the timescale during which

measurements were made, the steady-state assumption seems reasonable and we continue to rely on it here.

Steady states can arise in one of two different ways, either at thermodynamic equilibrium or away from equilibrium. The profound difference between these situations is not always appreciated. We focus in this section on thermodynamic equilibrium and discuss later what happens away from equilibrium.

## Thermodynamic equilibrium and free energy

Suppose that we have a system, such as DNA and its associated molecular regulators like TFs, that is able to exchange both molecules and heat energy, by way of molecular collisions, with a surrounding buffer or 'heat bath'. Suppose further that the system and its buffer are contained within an isolated compartment, so that the total numbers of molecules and the total energy within that compartment are conserved. Physics tells us that the compartment will eventually reach a steady state, as described above, with both the system and the buffer at the same temperature and that, furthermore, this steady state of thermodynamic equilibrium will satisfy special properties, described below. For the system in *Figure 1B* to be treated in this way, the compartment must be isolated, so that neither TF is being synthesised or degraded over the timescale of interest and no external sources of matter or energy, such as ATP, are being continually used behind the scenes.

The consequences of thermodynamic equilibrium are most easily understood through a free-energy landscape, a concept that has been widely used to study protein dynamics (*Frauenfelder et al., 1991*). The system of binding sites on DNA is assumed to have a free energy that is a function of the constituent atomic motions and inter-atomic forces. These atomic features inhabit an enormously high-dimensional space but, if we were to pretend that it is only two-dimensional, then a hypothetical free-energy function for our system might resemble what is plotted in *Figure 1C*. We can imagine our system as a marble rolling over this surface under the influence of gravity, with the gravitational potential energy playing the role of the free energy, while being continually buffeted by random molecular collisions from the surrounding heat bath. Provided the buffeting is not too vigorous, the system will come to rest in a basin, whose lowest point is a local minimum of the free energy. The basins correspond to the microstates. They are the stable entities that emerge from the atomic interactions. Here, 'stable' means that, provided the marble is not pushed too far from the minimum, it remains within the basin.

Stable does not mean unchanging. If the marble acquires sufficient energy from the heat bath, it can get over the surrounding mountain range and reach another basin, leading to a transition between microstates. To do this, the marble has to cross the 'continental divide' that separates the two basins. The easiest way, in free-energy terms, is at the lowest point on the continental divide. If you imagine pouring water into the source basin, this lowest crossing point is found where the water first starts to flow into the target basin. A hypothetical trajectory connecting microstate **o** to microstate **so** through the lowest crossing point is shown in *Figure 1D*. This diagram will help us to interpret the labels in *Figure 1B* at thermodynamic equilibrium.

There is a particularly important relationship between the free-energy minima and the edge labels at thermodynamic equilibrium: the ratio of the binding label to the unbinding label is determined solely by the free-energy difference between the corresponding microstates. Specifically, for microstates **o** and **so** as illustrated in *Figure 1D* (and similarly elsewhere in the graph),

$$\frac{k_4^+[S]}{k_4^-} = \exp\left(\frac{\Delta\Phi}{k_BT}\right). \tag{3}$$

Here, $\Delta\Phi$ is the difference between the free-energy minima of microstates **o** and **so**, while $k_B$ is Boltzmann's constant and $T$ is the absolute temperature. (Recall that $k_BT$ sets the molecular energy scale: the average kinetic energy of a molecule of an ideal gas is $3k_BT/2$.) *Equation 3* is one of the basic equations of statistical mechanics and goes back to Boltzmann and Maxwell.

*Equation 3* tells us that the ratio of the binding to unbinding edge label is a thermodynamic quantity, in the sense that it is determined by just the free-energy minima. The individual edge labels are another matter. They depend not only on the free-energy minima but also on the height of the barrier between the minima at the lowest crossing of the continental divide. In *Figure 1D*, it is expected that the binding label, $k_4^+[S]$, depends on $\Delta\Phi^+$ and the unbinding label, $k_4^-$, depends on

$\Delta\Phi^-$. However, in contrast to *Equation 3*, the specific dependencies are not easily calculated. The individual edge labels are kinetic, not thermodynamic, quantities: they depend on the overall shape of the free-energy landscape and not just on its minima.

## Calculating $R$ at thermodynamic equilibrium

*Equation 3* allows us to understand the quantities $K_i$ and $R$ in *Equation 1*. In the treatment given by Chen et al, the $K_i$ in *Equation 1* are interpreted in terms of the model in *Figure 1B* as the ratios of the corresponding binding label to unbinding label (see the note above on the convention being used here). Specifically,

$$K_i = \frac{k_i^+ [X]}{k_i^-},$$ (4)

where $X = S$ for $i = 1, 4$ and $X = O$ for $i = 2, 3$. Note what happens when we multiply the $K$ quantities along the path in the graph from **e** to **so** that goes through **s**, to get $K_1 K_3$. Because of *Equation 3*, the free-energy differences within the exponential add together so that the result depends only on the difference in free energy between **e** and **so** (Materials and methods). In particular, it does not depend on having taken the path that goes through **s**. If we calculate the product along the path that goes through **o**, to get $K_2 K_4$, we must get the same result, so that $K_1 K_3 = K_2 K_4$. Hence, at thermodynamic equilibrium, $R$ in *Equation 1* always satisfies $R = 1$ (Materials and methods).

As explained above, Chen et al. assumed that $R = \rho$ and their data showed that $\rho = 3.05$. On this basis, we would have to conclude, as a matter of fundamental physics, that the system described by *Figure 1B* is being maintained away from thermodynamic equilibrium by some form of energy expenditure behind the scenes.

## Ordered assembly and $R$

Chen et al. did not infer the existence of energy expenditure. Instead, they took their calculation that $R = \rho = 3.05$ to imply an asymmetry between the probabilities of taking the two paths between **e** and **so**. The energy landscape in *Figure 1C* helps us understand how binding asymmetry can arise at thermodynamic equilibrium. Imagine that the mountain ridge that separates the basin of **o** from the basin of **so** becomes much higher, so that it gets much harder for the marble (ie: the system) to acquire energy from the heat bath and make it over the continental divide between microstate **o** and microstate **so**. Of course, for the same reason it will also be harder for the marble to go from **so** back to **o**. Notice that this kind of change can occur in such a way that $\Delta\Phi^+$ and $\Delta\Phi^-$ in *Figure 1D* become much larger, while $\Delta\Phi$ remains the same. This increased barrier will give rise to an asymmetry in binding order: our nanoscopic drone will see Sox2 binding after Oct4 much less frequently than Oct4 binding after Sox2. However, because $\Delta\Phi$ has not changed, there is no change in $K_4$ and hence no change in $R$, as given by *Equation 1*. We see that binding-order asymmetry is a kinetic quantity. In contrast, because of *Equation 1* and *Equation 3*, $R$ is a thermodynamic quantity. The two quantities are unrelated to each other.

It is not difficult to correctly calculate the binding-order asymmetry for the model in *Figure 1B*. The ratio of the steady-state probability of taking the upper path, from **e** to **so** through **s**, to the steady-state probability of taking the lower path, from **e** to **so** through **o**, is given by (Materials and methods),

$$\frac{k_1^+ k_3^+ \left(k_2^- + k_4^+ [S]\right)}{k_2^+ k_4^+ \left(k_1^- + k_3^+ [O]\right)}.$$ (5)

As might have been expected, the binding-order asymmetry depends on the concentrations of the TFs. In contrast, $R$ is independent of TF concentrations (*Equation 6*). Further analysis of binding-order asymmetry will need to take into account the more elaborate models introduced below and lies outside the scope of the present paper.

## The cycle condition and detailed balance

Thermodynamic equilibrium is a steady state with a very special property. This is most readily seen by substituting in *Equation 1* the expressions for $K_i$ in terms of the labels (*Equation 4*). The free concentrations $[O]$ and $[S]$ cancel out to give,

$$R = \frac{k_1^+ k_3^+ k_4^- k_2^-}{k_3^- k_1^- k_2^+ k_4^+} . \tag{6}$$

We see that $R$ is the product of the rate constants taken clockwise around the cycle in *Figure 1B*, divided by the product of the rate constants taken anti-clockwise. The requirement that $R = 1$ is the 'equilibrium cycle condition' for a linear framework graph (Materials and methods). If such a graph represents a system at thermodynamic equilibrium then every cycle in the graph satisfies the equilibrium cycle condition. This imposes constraints on the values of the edge labels. If a system can reach thermodynamic equilibrium, its on-rates and off-rates cannot vary independently but must satisfy the cycle condition. This is the special property that systems at thermodynamic equilibrium exhibit, beyond being at steady state.

The equilibrium cycle condition has a surprising consequence: each pair of binding and unbinding edges must be in balance, independently of any other edges in the graph (*Gunawardena, 2012*). This is the principle of 'detailed balance', first introduced into chemistry by Gilbert Lewis (*Lewis, 1925*). For example, if we let $\Pr(\mathbf{x})$ denote the steady-state probability of microstate $\mathbf{x}$ (here, $\mathbf{x} = \mathbf{e}, \mathbf{o}, \mathbf{s}$ or $\mathbf{so}$), then at thermodynamic equilibrium, considering the pair of edges between $\mathbf{e}$ and $\mathbf{s}$, the forward flux of probability from $\mathbf{e}$ to $\mathbf{s}$ is exactly balanced by the reverse flux from $\mathbf{s}$ to $\mathbf{e}$, so that

$$(k_1^+ [S]) \Pr(\mathbf{e}) = (k_1^-) \Pr(\mathbf{s}) . \tag{7}$$

Despite being connected in a cycle, *Equation 7* shows that the model in *Figure 1B* behaves as if it is four uncoupled pairs of transitions, which are linked only by the common probability distribution on the microstates. This remarkable uncoupling under detailed balance demonstrates the special nature of thermodynamic equilibrium.

At equilibrium, a flux balance equation like *Equation 7* holds for any pair of microstates that are linked by binding and unbinding edges. This yields a simple rule for calculating the equilibrium steady-state probability of any microstate. It is only necessary to take any contiguous path of binding and unbinding edges to that microstate from $\mathbf{e}$ and to multiply the label ratios along the path. For instance, taking the path through $\mathbf{s}$ and using *Equation 4*, we see that

$$\Pr(\mathbf{so}) = K_3 K_1 \Pr(\mathbf{e}) . \tag{8}$$

The equilibrium cycle condition, $R = 1$, ensures that the result in *Equation 8* does not depend on the chosen path. Since the total probability of all microstates is $1$, this allows the probability of each microstate to be determined in terms of the $K_i$ (see below for an example). By way of *Equation 3*, this prescription for calculating equilibrium steady-state probabilities becomes identical to that of equilibrium statistical mechanics (*Bintu et al., 2005*; *Segal and Widom, 2009*; *Sherman and Cohen, 2012*). It is important to note that *Equation 8* does not hold away from thermodynamic equilibrium, where quite different methods must be used to calculate steady-state probabilities (Materials and methods).

*Equation 8* shows why it is better to define quantities like $K_i$ as we have done, with the binding labels in the numerators. The steady-state probability of a microstate is then calculated by simply multiplying the $K_i$ along a path from the empty microstate $\mathbf{e}$. Had we used the same convention as Chen et al., we would have had to multiply reciprocals, which makes the formulas less transparent.

The classical thermodynamic principles underlying energy and entropy at equilibrium were first derived for macroscopic systems for which a microscopic description was not available (steam engines, for instance). For this reason, they remain among the most general physical laws known to us. Detailed balance is an additional property that holds at thermodynamic equilibrium, which presupposes a microscopic description and which cannot be derived from classical thermodynamics. We have explained here why detailed balance must hold if there is an underlying free-energy landscape. This offers helpful intuition but the landscape itself is a complex concept. Detailed balance

can be directly derived, without assuming an energy landscape, from the time-reversal symmetry of the physical laws that govern microscopic interactions (*Tolman, 1938*; *Mahan, 1975*). Indeed, the hallmark of thermodynamic equilibrium, as implied by *Equation 7*, is that we should not be able to tell whether the movie taken by our nanoscopic drone is running forwards or in reverse. This 'microscopic reversibility' illustrates again the unusual nature of thermodynamic equilibrium.

## Calculating $\rho$ in the single-locus model

With the biophysical foundation described above at our disposal, we can return to the analysis of the experimental context and data of Chen et al. We will first describe how Chen et al. measured the quantities $\kappa_i$ and then reconsider how they interpreted them in terms of the model in *Figure 1A*. Chen et al. used 2D single-molecule tracking to estimate the average residence time a TF remains bound to a specific binding site on DNA (~10 s) and 3D multifocus microscopy to estimate the average residence time a TF spends between specific binding sites (~400 s). *Figure 2A* shows the corresponding graph from the TF viewpoint. Here, molecules of a TF called $X$ are being observed, each of which is assumed to shuttle stochastically between two microstates. In microstate **b**, $X$ is bound to a DNA site that is specific for $X$ binding, while in microstate **nb**, $X$ is not bound to such a site. We regard the $X$-specific binding sites on DNA as a molecular species, called $D_X$, whose free concentration (i.e. the concentration of such sites which are not bound by $X$) appears in the binding label.

The microstate **nb** is potentially complicated as it may lump together non-specific binding to DNA, 1D diffusion along DNA and anomalous, as well as normal, 3D diffusion in the nuclear environment. These complications are discussed in *Chen et al. (2014)*. We do not analyze them further here and take *Figure 2A* to be the basic summary of the data of Chen et al., with the binding edge label, $k^+[D_X]$, and the unbinding edge label, $k^-$, being the reciprocals of the corresponding average residence times, measured as described above.

Only a single TF could be measured at one time by the methods in *Chen et al. (2014)*. To accommodate the other TF, Chen et al. undertook experiments in a stably transformed 3T3 cell line that constitutively expressed one TF, coupled to a fluorescent tag for single-molecule measurements, with the other TF under an inducible expression system. The 3T3 line does not normally express endogenous Sox2 or Oct4, enabling the recombinant proteins to be measured without background interference. Two cell lines were prepared, one in which Sox2 was measured and Oct4 was induced and the other in which Oct4 was measured and Sox2 was induced.

Chen et al. estimated the values $\kappa_1$, $\kappa_2$, $\kappa_3$ and $\kappa_4$ in *Equation 2* using four experimental scenarios as follows. In each case, the $\kappa_i$ values were calculated as

$$\kappa_i = \frac{k^+[D_X]}{k^-}, \tag{9}$$

where $k^+[D_X]$ is the binding edge label and $k^-$ is the unbinding edge label for the TF graph in *Figure 2A*, for the scenario in question (Materials and methods). Note that $k^+$ and $k^-$ differ between scenarios but we do not show this to avoid complicating the notation. $\kappa_1$ and $\kappa_4$ were determined using the cell line in which Sox2 was measured: for $\kappa_1$, Oct4 was not induced and for $\kappa_4$, Oct4 was maximally induced. $\kappa_2$ and $\kappa_3$ were determined in a similar way using the cell line in which Oct4 was measured: for $\kappa_2$, Sox2 was not induced and for $\kappa_3$, Sox2 was maximally induced.

As pointed out previously, there are difficulties with the interpretation made by Chen et al. of the $\kappa_i$ in terms of what happens on DNA. Most importantly, they assumed that the binding and unbinding labels in the TF graph in *Figure 2A* were 'good estimates' for the binding and unbinding labels, respectively, in the DNA graph in *Figure 1B*, (*Chen et al., 2014* page S7). This is correct for the unbinding labels: when the specifically bound state of TF and DNA breaks apart, the rate at which this happens is the same whether we follow the TF or follow the DNA. However, it is a different matter for the binding labels. When the TF $X$ moves from being not specifically bound to being specifically bound to DNA, its rate depends on the concentration of $X$-specific DNA-binding sites that are unoccupied, or $[D_X]$ in *Figure 2A*. In contrast, when an unoccupied DNA site is being specifically bound by $X$, the rate depends on the concentration of free TF molecules, or $[X]$ in *Figure 1B*. The corresponding binding labels are different.

A further difficulty arises because TF concentrations will differ depending on whether or not the TF is being induced. The model in *Figure 1B*, which is a rigorous version of *Figure 1A*, does not

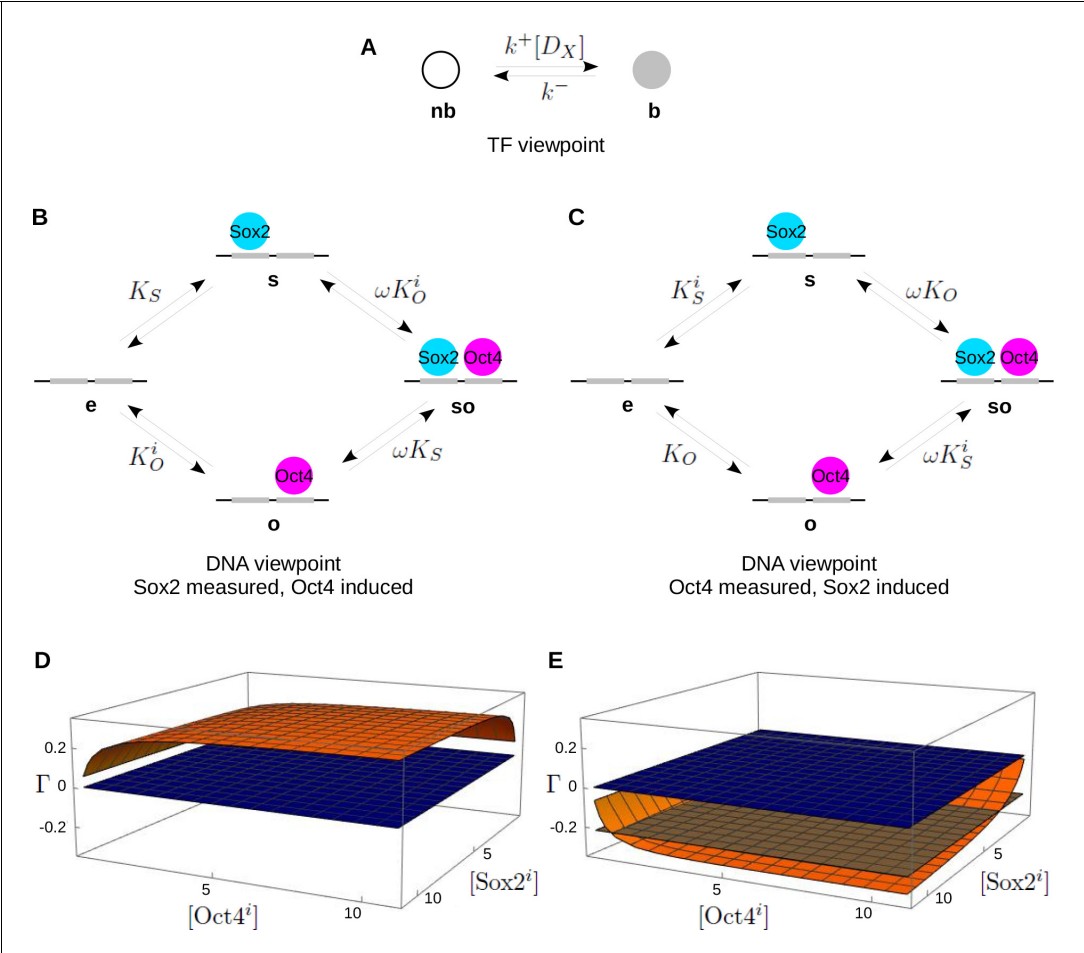

**Figure 2.** The single-locus model and reciprocity. (**A**) Linear framework graph for the TF $X$, showing the microstate, **b**, in which $X$ is specifically bound to DNA and the microstate, **nb**, in which $X$ is not specifically bound. (**B**) Graph adapted from *Figure 1B* for the cell line in which Sox2 is measured and Oct4 is induced and taken to be at thermodynamic equilibrium, with only the ratios of binding to unbinding labels being shown. The subscripts $S$ and $O$ denote Sox2 and Oct4, respectively, the superscript $i$ denotes 'induced' and $\omega$ is the cooperativity. (**C**) As in B for the cell line in which Oct4 is measured and Sox2 is induced. Panels B and C define the single-locus model at thermodynamic equilibrium. (**D**) Plot of the reciprocity, $\Gamma$, (orange surface), as defined in *Equation 20*, against the concentrations, $[\text{Sox2}^i]$ and $[\text{Oct4}^i]$, of the induced TFs, for the single-locus model with parameter values at thermodynamic equilibrium. The flat blue plane indicates $0$. (**E**) As in panel D but for parameter values away from thermodynamic equilibrium, calculated as described in the text and the Materials and methods. Only a single off-rate, corresponding to $k_4^-$ from **so** to **o** in *Figure 1B*, was increased from the equilibrium value used in panel D, thereby breaking detailed balance. With this change, the reciprocity becomes negative throughout and can equal the reciprocity of $\Gamma = -0.22$ calculated from the data of Chen et al. (Materials and methods), indicated by the flat brown plane. Numerical parameter values are given in the Materials and methods.

DOI: https://doi.org/10.7554/eLife.41017.004

account for this concentration difference and is, therefore, not an adequate description of the experimental context of Chen et al.

Let us first reformulate the graph in *Figure 1B* to reflect the experimental context and then explain how to move between the TF viewpoint and the DNA viewpoint. *Figure 2B and C* show the DNA microstate graphs corresponding to *Figure 1B* for the cell lines in which Sox2 is measured and Oct4 is induced and Oct4 is measured and Sox2 is induced, respectively. We have made the assumption that these models are at thermodynamic equilibrium and have simplified the graphs accordingly. Because of *Equation 4* and *Equation 8*, we need only keep track of the ratios of binding to unbinding labels, which we denote generically by $K_X$, where $X$ is either $S$ or $O$, depending on which TF is binding. We should keep in mind that $K_X$ depends on the concentration of $X$ (*Equation 4*), so that its value will differ depending on whether $X$ is constitutively expressed or maximally

induced. We use $K_X$ for the former and $K_X^i$ for the latter, with $i$ signifying 'induced'. We assume that induction of one TF does not change the expression level of the other TF, which seems reasonable. However, the binding of one TF can be altered if the other TF is already bound to DNA. This cooperativity is accounted for by a non-dimensional factor, $\omega$, which is concentration independent (Materials and methods). It is a consequence of being at thermodynamic equilibrium that the influence of the TFs on each other through cooperativity is symmetric, so that the same $\omega$ works for both TFs (Materials and methods). If $\omega > 1$, the TFs help each other to bind ('positive' cooperativity); if $\omega < 1$, they hinder each other ('negative' cooperativity); if $\omega = 1$, they are independent and neither influences the other's binding (Materials and methods). *Figure 2B and C* constitute the 'single-locus' model that we will analyse in this section.

To understand the relationship between the TF microstate graph (*Figure 2A*) and the DNA microstate graph (*Figure 2B or C*), it is helpful to think of these graphs as arising from two different views of a hypothetical bi-molecular reaction

$$X + D_X \underset{k^-}{\overset{k^+}{\rightleftharpoons}} DX. \tag{10}$$

Here, TF molecules that are not specifically bound, denoted $X$, interact with DNA sites that are specific for $X$ binding, $D_X$, to form the bound combination, $DX$.

The TF microstate graph arises from *Equation 10* by focussing on $X$ and $DX$, which correspond to the microstates **nb** and **b**, respectively, in *Figure 2A*. In contrast, the DNA microstate graphs (*Figure 2B and C*) arise by focussing on $D_X$ and $DX$ but in a more complicated way because of the presence of the other TF. The link between the TF and DNA graphs comes through $DX$, which is common to both viewpoints. We will calculate the steady-state concentration of $DX$ in two ways, using the TF microstate graph and the DNA microstate graph, and equate the results. This will allow us to interpret the $\kappa_i$, which were calculated from the TF graph in *Figure 2A*, in terms of the quantities in the DNA graphs in *Figure 2B and C*.

This strategy for moving between TF microstates, which are directly observed, and DNA microstates, which are not, may be broadly useful for other single-molecule studies.

Let us first calculate $[DX]$ from the TF graph viewpoint (*Figure 2A*). There, $DX$ corresponds to the specifically bound microstate **b**. We can calculate the steady-state probability of **b** in two ways. On the one hand, by detailed balance (*Equation 7*), $\mathrm{Pr}(\mathbf{b}) = \kappa \, \mathrm{Pr}(\mathbf{nb})$, where we have used $\kappa$ as a generic symbol for the quantity defined in *Equation 9*; we will decide which $\kappa_i$ is involved below, depending on which cell line we are considering. Since $\mathrm{Pr}(\mathbf{b}) + \mathrm{Pr}(\mathbf{nb}) = 1$, we find that $\mathrm{Pr}(\mathbf{b}) = \kappa/(1 + \kappa)$. On the other hand, $\mathrm{Pr}(\mathbf{b})$ can also be interpreted at thermodynamic equilibrium in a different way. Since each $X$ molecule is either specifically bound, in microstate **b**, or not specifically bound, in microstate **nb**, $\mathrm{Pr}(\mathbf{b})$ is just the fraction of $X$ molecules that are specifically bound. If we denote the concentration of total TF by $[X]_{\text{tot}}$, so that $[DX] + [X] = [X]_{\text{tot}}$, then $\mathrm{Pr}(\mathbf{b}) = [DX]/[X]_{\text{tot}}$. It follows that

$$\frac{[DX]}{[X]_{tot}} = \mathrm{Pr}(\mathbf{b}) = \frac{\kappa}{1 + \kappa}. \tag{11}$$

Now let us calculate $[DX]$ from the DNA graph viewpoint. There are four scenarios to consider, as described above. Under induction of the non-measured TF, we expect the rate constants in *Equation 10* to change but the total concentration of the measured TF, given by $[X]_{tot} = [X] + [DX]$, and the total concentration of $X$-specific binding sites, given by $[D_X]_{tot} = [D_X] + [DX]$, not to change. The quantity $[DX]/[D_X]_{tot}$ is then the fraction of $X$-specific sites that are bound by $X$ at steady state. This fraction can be calculated from the DNA graph that is appropriate for the scenario being considered and it has the general form $B/(A + B)$. Here, $A$ and $B$ are terms that depend on the particular scenario but have a straightforward interpretation: up to a scalar multiple, $A$ is the sum of the probabilities of microstates in which the measured TF is not bound and $B$ is the sum of the probabilities of microstates in which the measured TF is bound. Hence,

$$\frac{[DX]}{[D_X]_{tot}} = \frac{B}{A + B}. \tag{12}$$

We now have two expressions for the common quantity $[DX]$. Letting $\alpha_X = [D_X]_{tot}/[X]_{tot}$ and equating $[DX]$ in *Equations 11 and 12*, we find that,

$$\kappa = \frac{B\alpha_X}{A + B(1 - \alpha_X)},$$ (13)

which describes how the quantities in the TF graph (*Figure 2A*) and the DNA graphs (*Figure 2B and C*) are related. The quantity $\alpha_X$, which involves the total concentrations of TF $X$ and of specific binding sites for $X$, can change with the experimental conditions and is generally unknown.

We can now calculate the $\kappa_i$ and thereby determine $\rho$ using *Equation 2*. It will be more informative, however, to separately calculate the ratios $\kappa_4/\kappa_1$ and $\kappa_3/\kappa_2$. These ratios each arise from one of the two cell lines and determine in each case the impact of induction of one TF on the other. $\rho$ is then easily found by recalling from *Equation 2* that,

$$\rho = \left(\frac{\kappa_4}{\kappa_1}\right)^{-1} \left(\frac{\kappa_3}{\kappa_2}\right).$$ (14)

To determine $\kappa_4$, consider the scenario for *Figure 2B* when Oct4 is maximally induced. $\kappa$ in *Equation 11* then corresponds to $\kappa_4$. This is where we run into a further problem with the analysis made by Chen et al. They assumed that, under maximal induction, all sites of specific Oct4 binding were occupied at steady state. This is unlikely to be the case, as other studies of TF binding have shown (*Morisaki et al., 2014*). Instead, we should allow for the possibility that Sox2 will encounter the microstate in which Oct4 is absent, **e**, as well as the microstate in which it is present, **o**. In this case, $DX$ in *Equation 10*—here, $DS$—corresponds to a combination of microstates **s** and **so**.

We can use the prescription in *Equation 8* to calculate the steady-state probabilities of individual microstates at thermodynamic equilibrium (Materials and methods). It is then straightforward to determine the fraction of sites that are bound by Sox2 as an average over these probabilities. We find that (Materials and methods),

$$\frac{[DS]}{[D_S]_{tot}} = \frac{K_S(1 + \omega K_O^i)}{1 + K_O^i + K_S(1 + \omega K_O^i)}.$$

In terms of *Equation 12*, $B = K_S(1 + \omega K_O^i)$ and $A = 1 + K_O^i$. We can now use *Equation 13*, remembering that in this scenario $X = S$ and $\kappa = \kappa_4$, to get

$$\kappa_4 = \frac{K_S(1 + \omega K_O^i)\alpha_S}{1 + K_O^i + K_S(1 + \omega K_O^i)(1 - \alpha_S)}.$$

To determine $\kappa_1$ is simpler because Oct4 is not induced. In this scenario, the DNA graph reduces to the microstates **e** and **s** in *Figure 2B*. This is analogous to the calculation which led to *Equation 11* for the TF viewpoint and we see by a similar argument that,

$$\frac{[DS]}{[D_S]_{tot}} = \frac{K_S}{1 + K_S}.$$

It follows from *Equation 13* that,

$$\kappa_1 = \frac{K_S\alpha_S}{1 + K_S(1 - \alpha_S)}.$$

Putting together the calculations for $\kappa_4$ and $\kappa_1$, we find that,

$$\frac{\kappa_4}{\kappa_1} = \frac{1 + \omega K_O^i + K_S(1 + \omega K_O^i)(1 - \alpha_S)}{1 + K_O^i + K_S(1 + \omega K_O^i)(1 - \alpha_S)},$$ (15)

and a similar calculation of $\kappa_3$ and $\kappa_2$ using *Figure 2C* shows that,

$$\frac{\kappa_3}{\kappa_2} = \frac{1 + \omega K_S^i + K_O(1 + \omega K_S^i)(1 - \alpha_O)}{1 + K_S^i + K_O(1 + \omega K_S^i)(1 - \alpha_O)}.$$ (16)

In each of the expressions in *Equations 15 and 16*, the only difference between the numerator and

the denominator lies in the second term, $K_O^i$ and $K_S^i$, respectively, which is multiplied by $\omega$ in the numerator. If $\omega = 1$, so that the TFs are independent, then $\kappa_4/\kappa_1 = \kappa_3/\kappa_2 = 1$. It follows from *Equation 14* that, in this case, $\rho = 1$. However, TFs that work in concert are expected to exhibit cooperativity in vivo (*Courey, 2001*) and in-vitro measurements show that Sox2 and Oct4 have substantial positive cooperativity on canonical binding motifs (*Ng et al., 2012*). If $\omega \neq 1$, then $\kappa_4/\kappa_1 \neq 1$ and $\kappa_3/\kappa_2 \neq 1$. It follows from *Equation 14* that $\rho \neq 1$, except for the implausible coincidence in which $\kappa_4/\kappa_1 = \kappa_3/\kappa_2$, which may be discounted. Hence, in answer to the question raised in the Introduction, we can be fooled into thinking that the model in *Figure 1A* is away from thermodynamic equilibrium by the conflation of $R$ with $\rho$ made by Chen et al. For the single-locus model in *Figure 2B and C*, the estimate that $\rho = 3.05$ provides no evidence for energy expenditure away from thermodynamic equilibrium.

## Reciprocity and evidence for energy expenditure

There is, however, more information to be gleaned about the data of Chen et al. from the analysis above. We note that the cooperativity, $\omega$, has the same value in both cell lines. It reflects the inherent concentration-independent influence of the TFs upon each other at thermodynamic equilibrium and it is the one parameter that is common to both *Figure 2B and C*. It follows from *Equations 15 and 16* that, at thermodynamic equilibrium, we must have either

$$\underbrace{\frac{\kappa_4}{\kappa_1} > 1 \,,\, \frac{\kappa_3}{\kappa_2} > 1}_{\text{when } \omega > 1} \quad \text{or} \quad \underbrace{\frac{\kappa_4}{\kappa_1} < 1 \,,\, \frac{\kappa_3}{\kappa_2} < 1}_{\text{when } \omega < 1} . \tag{17}$$

However, according to the data of Chen et al. (Materials and methods),

$$\frac{\kappa_4}{\kappa_1} = 0.86 \,,\, \frac{\kappa_3}{\kappa_2} = 2.57 . \tag{18}$$

This arrangement of values on either side of 1 is inconsistent with *Equation 17*. (We will address the statistical significance of this estimate below.) *Equation 18* cannot be accounted for by the single-locus model at thermodynamic equilibrium. Accordingly, it appears that the data of Chen et al. do suggest energy expenditure away from thermodynamic equilibrium. Moreover, the condition in *Equation 17* contains more information relevant to assessing this than does the value of $\rho$.

This suggests a more productive way to exploit the data and experimental methodology of Chen et al. Let us first note that we are less interested in the $\kappa_i$ than in quantities which measure TF binding to DNA. Accordingly, let $P_i$ denote the fraction of $X$-specific DNA-binding sites that are bound by $X$ at steady state, or $[DX]/[D_X]_{tot}$, where $i$ indexes the same scenario that gives $\kappa_i$. We can determine $P_i$ from the corresponding DNA graph by using *Equation 12*. Since $\alpha_X = [D_X]_{tot}/[X]_{tot}$, it also follows from *Equation 11* that,

$$\alpha_X P_i = \frac{\kappa_i}{1 + \kappa_i} , \tag{19}$$

which makes it straightforward to calculate $P_i$ from $\kappa_i$ and $\alpha_X$.

Since $P_4$ and $P_1$ both require $\alpha_S$ in *Equation 19* while $P_3$ and $P_2$ both require $\alpha_O$, the ratios $P_4/P_1$ and $P_3/P_2$, in which the unknown $\alpha_X$'s cancel out, offer a proxy for the ratios $\kappa_4/\kappa_1$ and $\kappa_3/\kappa_2$, respectively. That is, using *Equation 19*, $P_4/P_1 > 1$ or $P_4/P_1 < 1$, if, and only if, $\kappa_4/\kappa_1 > 1$ or $\kappa_4/\kappa_1 < 1$, respectively, and similarly for $P_3/P_2$ and $\kappa_3/\kappa_2$. We can exploit this proxy relationship to introduce the following quantity, $\Gamma$, which we call the 'reciprocity' of the two TFs,

$$\Gamma = \left( \frac{P_4}{P_1} - 1 \right) \left( \frac{P_3}{P_2} - 1 \right) . \tag{20}$$

Because of the proxy relationship between the $\kappa_i$ ratios and the $P_i$ ratios, if there is either positive cooperativity, with $\omega > 1$, or negative cooperativity, with $\omega < 1$, the reciprocity $\Gamma$ remains positive, with $\Gamma > 0$. The reciprocity thereby provides a single number as a succinct summary of *Equation 17*. But the reciprocity also has a meaningful interpretation in terms of TF binding to DNA. If $\Gamma > 0$, then either the fractional bindings of both TFs increase with increased expression of the other TF, so that

$P_4/P_1 > 1$ and $P_3/P_2 > 1$, or both fractional bindings decrease, so that $P_4/P_1 < 1$ and $P_3/P_2 < 1$. With positive reciprocity, the TFs must influence each other's fractional binding in the same direction.

It follows from *Equation 17*, using the proxy relationship between the $\kappa_i$ ratios and the $P_i$ ratios, that the single-locus model at thermodynamic equilibrium exhibits positive reciprocity, with $\Gamma > 0$. In this case, the sign of the reciprocity does not depend on the quantities in *Figure 2B and C*, although the specific numerical value of $\Gamma$ may do so. The sign is also much less sensitive to measurement errors than the value. *Figure 2D* shows a plot of $\Gamma$ as a function of the induced concentrations of Sox2 and Oct4, for the single-locus model at thermodynamic equilibrium; as expected, $\Gamma > 0$ throughout. Recall that $[\text{Sox2}^i]$ and $[\text{Oct4}^i]$ are the concentration factors in the quantities $K_S^i$ and $K_O^i$, respectively, which, unlike the on- and off-rates, would be expected to change with the experimental conditions.

The data of Chen et al. show that Sox2 and Oct4 exhibit negative reciprocity: using *Equations 19 and 20*, we find that $\Gamma = -0.22$ (Materials and methods). The statistical significance that $\Gamma < 0$ is not as strong as that for $\rho > 1$: the probability that $\Gamma \geq 0$ is $2 \times 10^{-2}$ (Materials and methods). However, this is a conservative estimate, which we made by assuming that the standard deviations reported by Chen et al. were based on the minimum of three samples (Materials and methods). If more samples were used, the significance would be higher. Notwithstanding that, the significance is still high enough to warrant further analysis.

*Equation 18* tells us that the fractional binding of Sox2 decreases when Oct4 expression is increased but the fractional binding of Oct4 increases when Sox2 expression is increased. This cannot occur for the single-locus model at thermodynamic equilibrium but *Figure 2E* shows that this level of negative reciprocity can be found away from equilibrium. The conditions under which the reciprocity becomes negative are not readily determined and depend on all the parameters.

To undertake the calculation in *Figure 2E*, the prescription for equilibrium steady-state probabilities in *Equation 8* can no longer be used and the equilibrium parameters in *Figure 2B and C* must be replaced with the individual labels, as in *Figure 1B*. One of the further advantages of the linear framework is that steady-state probabilities can still be calculated analytically away from equilibrium. The procedure has been detailed in *Estrada et al. (2016)* for a similar kind of graph to those in *Figure 2B and C* and is briefly reviewed in the Materials and methods. We conclude from the single-locus model that, although $\rho$ is not informative, the occurrence of negative reciprocity for the data of Chen et al. does provide evidence for energy expenditure away from thermodynamic equilibrium.

## Genomic diversity and mixed cooperativity

The single-locus model accommodates the experimental context but, as pointed out in the Introduction, it remains unclear how well it summarizes, in some average sense, how Sox2 and Oct4 bind to the genome. In reality, there are many loci at which these TFs bind, with potentially widely varying $K$'s and $\omega$'s. To understand how such diversity could affect the conclusion reached above, we considered the more general 'genomic-diversity' model in *Figure 3A*. This model allows for two types of genomic loci on DNA, type 1 and type 2, in proportions $l$ and $1 - l$, respectively, with each locus following the single-locus model. The $K$'s and $\omega$'s are the same at all loci of the same type, as indicated by the respective subscript, but can be different between the two types of loci. Of course, actual genomic diversity is much richer but the model in *Figure 3A* allows diversity to be accommodated in a tractable way.

From the TF viewpoint, the graph remains the same as in *Figure 2A*. From the DNA viewpoint, the graphs for each cell line must now include the many genomic loci present and can be very complicated. If there are $N$ loci overall, then the total number of microstates across the genome is $4^N$. To simplify the calculations, we assume that the loci are independent of each other, so that binding at one locus does not affect the binding at any other locus, whether of the same or of different type. In other words, we assume that there is no cooperativity whatsoever between sites in different loci. Higher-order cooperativity, through which multiple sites can influence binding at another site, is thought to be necessary to account for information integration in eukaryotic genomes (*Estrada et al., 2016*) but such quantities have yet to be measured. In the absence of data for higher order effects between Sox2 and Oct4 and in order to keep the calculation manageable, we allow only 'pairwise' cooperativity within a single locus, as described by $\omega$.

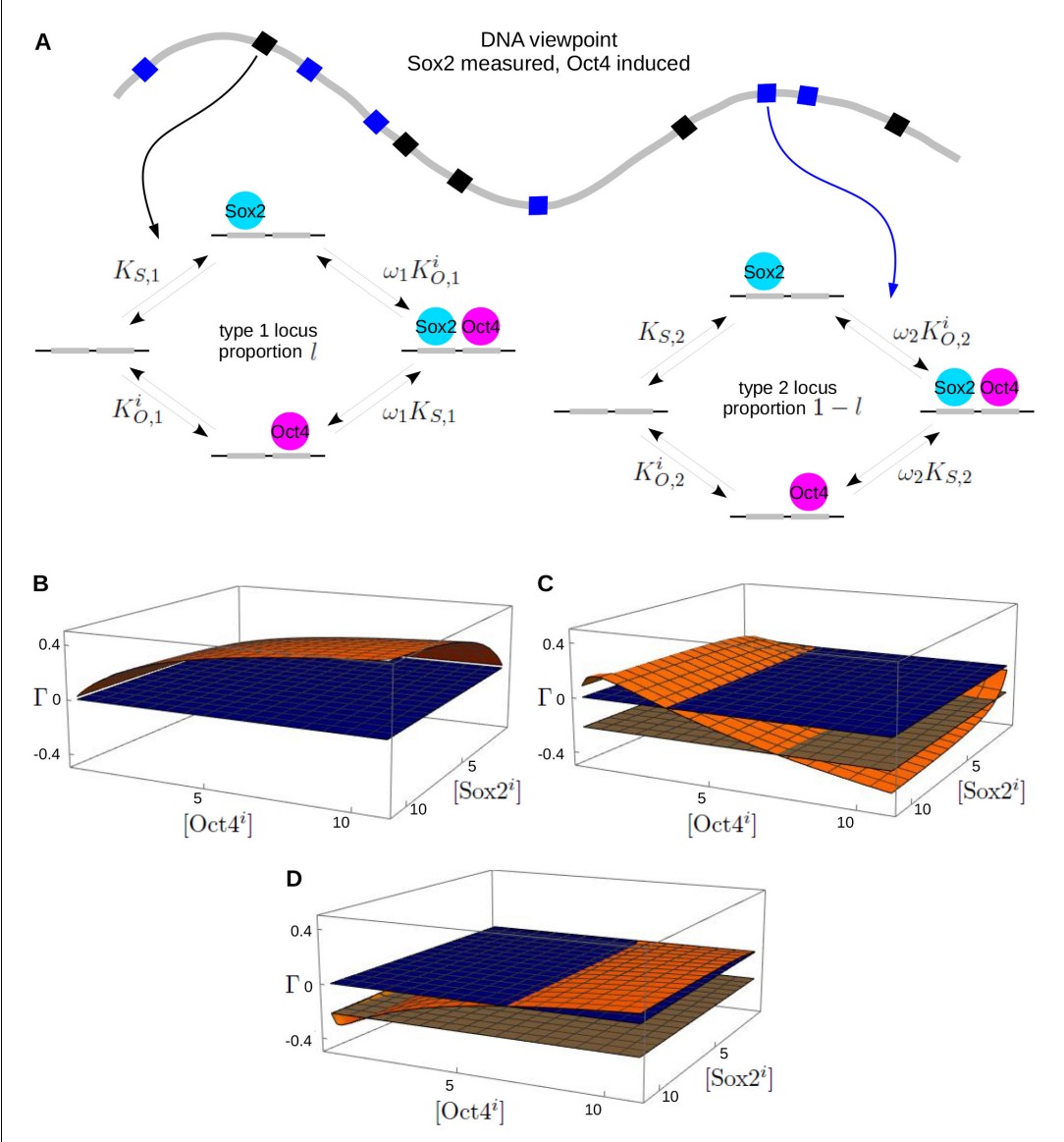

**Figure 3.** The genomic-diversity model and reciprocity. (A) Schematic illustration of diversity in Sox2 and Oct4 binding sites on genomic DNA (thick gray curve), showing two types of loci, type 1 (black squares) and type 2 (blue squares), which have distinct $K$'s and $\omega$'s, as indicated by the subscripts, 1 and 2, respectively. The details are shown for the cell line in which Sox2 is measured and Oct4 is induced, as in *Figure 2B* and a similar model should be imagined for the other cell line, as in *Figure 2C*. (B) Plot of the reciprocity, $\Gamma$, as in *Figure 2B*, for the model in panel A at thermodynamic equilibrium with both types of loci having positive cooperativity. (C) As in panel B but with the off-rate corresponding to $k_4^-$, for the transition from **so** to **o** in *Figure 1B*, increased at type II loci from the equilibrium value used in panel B, thereby breaking detailed balance. The flat brown plane marks the reciprocity, $\Gamma = -0.22$, of the data of Chen et al. (Materials and methods). (D) As in panels B and C but at thermodynamic equilibrium with mixed cooperativities, positive at type 1 loci and negative at type 2 loci. Numerical parameter values are given in the Materials and methods.
DOI: https://doi.org/10.7554/eLife.41017.005

If the loci are independent then the steady-state fraction of sites bound by a TF can be calculated as an average over the individual loci, irrespective of whether or not the model is at thermodynamic equilibrium (Materials and methods). If TF $X$ is being measured, then,

$$\frac{[DX]}{[D_X]_{tot}} = \underbrace{\text{fraction}\, D_X\, \text{bound}}_{\text{genome}} = l\underbrace{(\text{fraction}\, D_X\, \text{bound})}_{\text{type 1 locus}} + (1-l)\underbrace{(\text{fraction}\, D_X\, \text{bound})}_{\text{type 2 locus}}.$$

Here, by 'fraction $D_X$ bound' we mean the fraction of $X$-specific DNA-binding sites that are bound

by $X$ at steady state. These fractions can be calculated for type 1 and type 2 loci as above. Once again, there are four scenarios depending on the cell line and on whether the TF which is not measured is induced or not. The quantities $P_i$ can then be calculated, as they were for the single-locus model above, and the reciprocity, $\Gamma$, determined from *Equation 20* (Materials and methods).

We find that, if both type 1 and type 2 loci exhibit the same sign of cooperativity, so that either $\omega_1, \omega_2 > 1$ or $\omega_1, \omega_2 < 1$, then the reciprocity remains positive (Materials and methods), as illustrated in *Figure 3B*. However, negative reciprocity can now arise in one of two ways. First, if the model is away from thermodynamic equilibrium, then the reciprocity can become negative to the level of the data of Chen et al. (*Figure 3C*). Second, if the model remains at thermodynamic equilibrium but the two types of loci exhibit mixed cooperativity, so that $\omega_1 > 1, \omega_2 < 1$ or $\omega_1 < 1, \omega_2 > 1$, then the reciprocity can also become negative to the level of the data of Chen et al. (*Figure 3D*).

In vitro measurements show mixed cooperativities between Sox2 and Oct4 on a wide range of naked DNA sequences (*Chang et al., 2017*). It is difficult to assess how much this tells us about the in-vivo context being modelled here. As mentioned above, the models used here are coarse-grained and have effective parameters, so that the cooperativity, $\omega$, could arise indirectly through interaction of TFs with nucleosomes and co-regulators (*Mirny, 2010*; *Estrada et al., 2016*). At present, we have little idea of the range of such effective in-vivo cooperativities. Furthermore, as we see from *Figure 3D*, it is not just the existence of mixed cooperativities but the values of all the parameters that determine the sign of the reciprocity. It may be reasonable to expect both positive and negative cooperativity between Sox2 and Oct4 in vivo but it is hard to know whether this is sufficient to account for the negative reciprocity found by Chen et al.

In the absence of convincing data, we are left with two non-exclusive possibilities to account for the data of Chen et al. using the genomic-diversity model in *Figure 3A*. Either there is energy expenditure behind the scenes that maintains loci at which Sox2 and Oct4 bind away from thermodynamic equilibrium or, if not, there is mixed cooperativity across the genome, with Sox2 and Oct4 helping each other through positive cooperativity at some loci and hindering each other through negative cooperativity at other loci.

## Discussion

Chen et al. concluded in their paper that Sox2 and Oct4 exhibit ordered assembly, with 'Sox2 engaging the target DNA first, followed by assisted binding of Oct4' (*Chen et al., 2014*). As we have seen, this conclusion is not justified by their analysis and the question of binding order remains unresolved. Furthermore, the quantity $R$ which they calculated, although important as a measure of the cycle condition for the model in *Figure 1A*, turns out to be less informative about their data than the reciprocity introduced in *Equation 20*. Reciprocity is a quantitative measure of how two TFs influence each other's genomic binding upon induction of expression: binding changes in the same direction with positive reciprocity and in the opposite direction with negative reciprocity.

The experimental strategy of Chen et al., of measuring each TF in turn while inducing the other, can now be seen as a protocol for estimating the reciprocity of any two TFs. Aside from the single-molecule experiments themselves, this protocol involves two stable cell lines each with an inducible expression system. The 3T3 cells were also chosen because they do not express endogenous Sox2 and Oct4. However, it may be possible to account for endogenous TF expression by titrating the induced exogenous TF rather than just maximally inducing it. In effect, this would begin to map out the reciprocity surfaces plotted in *Figure 2D and E* and *Figure 3B, C and D*, whose shapes may reveal more about the impact of endogenous TFs. Indeed, much more may be learned by going beyond the simple positive or negative dichotomy on which we have focussed here.

What we have found in analyzing the sign of the reciprocity is quite different from ordered assembly. Far from Sox2 'assisting' Oct4, the data of Chen et al. may be accounted for, by the genomic-diversity model at thermodynamic equilibrium (*Figure 3A*), if Sox2 and Oct4 sometimes help and sometimes hinder each other across the genome. This finding was unexpected. It is not surprising that TFs like Sox2 and Oct4 exhibit different cooperativities at different loci. However, for the variation in sign of that cooperativity across the genome to be relevant has significant ramifications. It tells us that single-molecule data, when coupled to the experimental strategy of Chen et al., provides more information than one might have expected. Indeed, it is remarkable that a single number, $\Gamma = -0.22$, can suggest that Sox2 and Oct4 exhibit both positive and negative cooperativity

across genomic loci in vivo. However, by the same token, we may need to know a good deal more about how TFs interact across the genome to properly interpret single-molecule data. The implications of this finding will need to be addressed in future studies.

In our view, the most intriguing consequence of our analysis is the tantalizing possibility of energy expenditure in the action of Sox2 and Oct4 on DNA. Although much is known about the molecular mechanisms of energy transduction, they have for the most part been treated no differently from all the other molecular complexity that has been uncovered in eukaryotic genomes. As we pointed out in the Introduction, physics offers a more fundamental perspective, in which energy expenditure is crucial for realising functionality that could not be achieved without it (*Hopfield, 1974*; *Coulon et al., 2013*; *Estrada et al., 2016*; *Sharma and O'Brien, 2018*). Following the energy may offer an organizing principle that can help us rise above molecular complexity and understand the genomic logic behind evolutionary tinkering.

It is challenging, however, to be confident of energy expenditure in vivo, especially with indirect experimental methods which average over genomic loci. The most compelling demonstration of non-equilibrium behavior would come from directly measuring the loss of detailed balance (*Battle et al., 2016*): the movie taken by our nanoscopic drone should look different when played in reverse. Methods for imaging TFs at a specific genomic locus offer promise for achieving this. It is worth emphasising that the non-equilibrium regime requires very different ways of thinking, not to mention of calculation (Materials and methods), which have barely penetrated the study of gene regulation (*Ahsendorf et al., 2014*; *Scholes et al., 2017*; *Estrada et al., 2016*).

As we have seen, when single-molecule data are rigorously analyzed with the appropriate biophysical model, a great deal may be learned about the mechanisms acting 'behind the scenes'. However, it is important to keep in mind that models are like any other tool. They do some things well and one has to use them appropriately. The reality within a cell is not described by any model; at best, a model accurately describes our assumptions about that reality (*Gunawardena, 2014a*). The resulting conclusions are always contingent on those assumptions, a point usually lost in the widespread fixation with 'predictive models'. The analysis undertaken here illustrates the point. The single-locus model (*Figure 2B and C*) predicts that the action of Sox2 and Oct4 requires energy expenditure away from thermodynamic equilibrium but the genomic-diversity model (*Figure 3A*) offers a different explanation for the same data.

But why stop there? There are many other features that could be included in a model. For instance, much attention has been drawn to the existence in cells of non-membrane bounded, phase-separated, liquid compartments (*Hyman et al., 2014*). These could influence gene regulation by creating local concentrations of regulators within the nucleus. Such heterogeneity in the concentration of a TF, $X$, could be accommodated, however, by changes between $K_{X,1}$ and $K_{X,2}$ in the genomic-diversity model (*Figure 3A*). We feel, therefore, that phase-separation is not so likely to change our conclusions.

Higher order cooperativity is another matter. Our results show that pairwise cooperativity does affect our conclusions, so it is plausible that higher-order cooperativity does too. Indeed, it may have a more pronounced impact (*Estrada et al., 2016*). At present, we know little about such higher-order effects, other than that they appear essential and plausible on theoretical grounds (*Estrada et al., 2016*). It may be reasonable, therefore, not to consider them at this time but we should remain aware that they may subsequently turn out to be important. The same may be true for other known features that we have judged not to be relevant, not to mention the 'unknown unknowns' that lie beyond our imagination.

Evidently, we must draw the line somewhere in formulating a model. The choice of what assumptions to make is a matter of collective knowledge and personal judgement. It is crucial to make these underlying assumptions visible and to acknowledge that the conclusions drawn are contingent upon them and may be subject to change as we learn more. We conclude that experiment and theory will have to work hand in hand if we are to figure out how biology really works (*Phillips, 2015*; *Goldstein, 2018*).

## Materials and methods

### Statistical significance of $\rho$ and $\Gamma$

In the main text, we reported the statistical significance of $\rho > 1$ and $\Gamma < 0$. We explain here how we calculated these values. Both $\rho$ and $\Gamma$ are defined in terms of the quantities $\kappa_i$. For the former, the definition is given by *Equation 2*; for the latter, it follows from *Equations 19 and 20* that,

$$\Gamma = \left( \frac{\kappa_4(1+\kappa_1)}{\kappa_1(1+\kappa_4)} - 1 \right)\left( \frac{\kappa_3(1+\kappa_2)}{\kappa_2(1+\kappa_3)} - 1 \right). \tag{21}$$

The $\kappa_i$ are defined by *Equation 9*, as the ratio of the binding label to the unbinding label for the TF graph in *Figure 2A*, as appropriate for each of the four experimental scenarios. These labels are rates, with dimensions of $(\text{time})^{-1}$. Chen et al. estimated the corresponding reciprocal quantities— the average residence times—from their single-molecule measurements and report the resulting values in graphical form (*Chen et al., 2014*, Figure 5) and in numerical form in a Supplementary spreadsheet. We extracted from this spreadsheet the residence times relevant to us—Imaging Summary, rows 7–10 ('NIH 3T3 cells'), column I for the binding label ('Specific target search time') and column C for the unbinding label ('Long-lived bound residence time')—as listed below, along with the corresponding value of $\kappa_i$.

| $i$ | (binding label)$^{-1}$ | (unbinding label)$^{-1}$ | $\kappa_i$ |
|---|---|---|---|
| 1 | $274.7 \pm 17.4$ | $11.6 \pm 0.78$ | 0.042 |
| 2 | $366.6 \pm 12.1$ | $7.75 \pm 0.34$ | 0.021 |
| 3 | $158.9 \pm 11.2$ | $8.57 \pm 1.18$ | 0.054 |
| 4 | $397.0 \pm 35.8$ | $14.1 \pm 1.06$ | 0.036 |

Each residence time was given as mean in seconds plus or minus the standard deviation (SD) (*Chen et al., 2014* , Figure 5). As we want to know the true mean value, we use the standard error of the mean (SEM), which is defined by $\text{SEM} = \text{SD}/\sqrt{N}$, where $N$ is the number of samples. We were unable to determine from *Chen et al. (2014)* the number of samples acquired for the experiments in the 3T3 cell lines, so we took the conservative view that only the minimum number of $N = 3$ samples had been used. If the actual number of samples was larger, then the significance values reported below would be correspondingly higher.

We calculated an empirical distribution for $\rho$ and for $\Gamma$ by choosing each residence time independently from a normal distribution, with mean given by the values shown in the table above and standard deviation given by the corresponding SEM for $N = 3$ samples, and calculating the resulting values of $\rho$ and $\Gamma$ using *Equation 2* and *Equation 21*, respectively. We repeated this $10^7$ times to build up the empirical probability density functions for $\rho$ and for $\Gamma$, as shown in *Scheme 1*.

We then estimated the fraction of these distributions corresponding to $\rho \leq 1$ and $\Gamma \geq 0$. For $\rho$ we found no instances in which $\rho \leq 1$ and concluded after further calculations that its probability is less than $10^{-9}$. For $\Gamma$ we found that the probability that $\Gamma \geq 0$ was $2.36 \times 10^{-2}$. This value had converged, to the three significant figures shown, by $10^6$ calculations.

### Thermodynamic equilibrium

We provide here explanations for some of the statements made in the main text about thermodynamic equilibrium using the linear framework. Further details about the material in this section can be found in *Ahsendorf et al. (2014)*; *Estrada et al. (2016)*; *Gunawardena, 2012*; *Wong et al. (2018)*.

In the linear framework a microscopic system, such as that shown in *Figure 1A*, is described by a labeled, directed graph (hereafter, 'graph'), in which the vertices represent microstates, the edges represent transitions between microstates and the edge labels represent the transition rates with dimensions of $(\text{time})^{-1}$ (*Figure 1B*).

It is helpful to introduce some notation to describe graphs. Let $G$ be a graph. Its vertices will be denoted by the indices $1, \cdots, N$ and an edge from vertex $i$ to vertex $j$ by $i \to j$. The set of all vertices will be denoted $\nu(G) = \{1, \cdots, N\}$. To specify the edge label, we will write $i \xrightarrow{a} j$ or say that

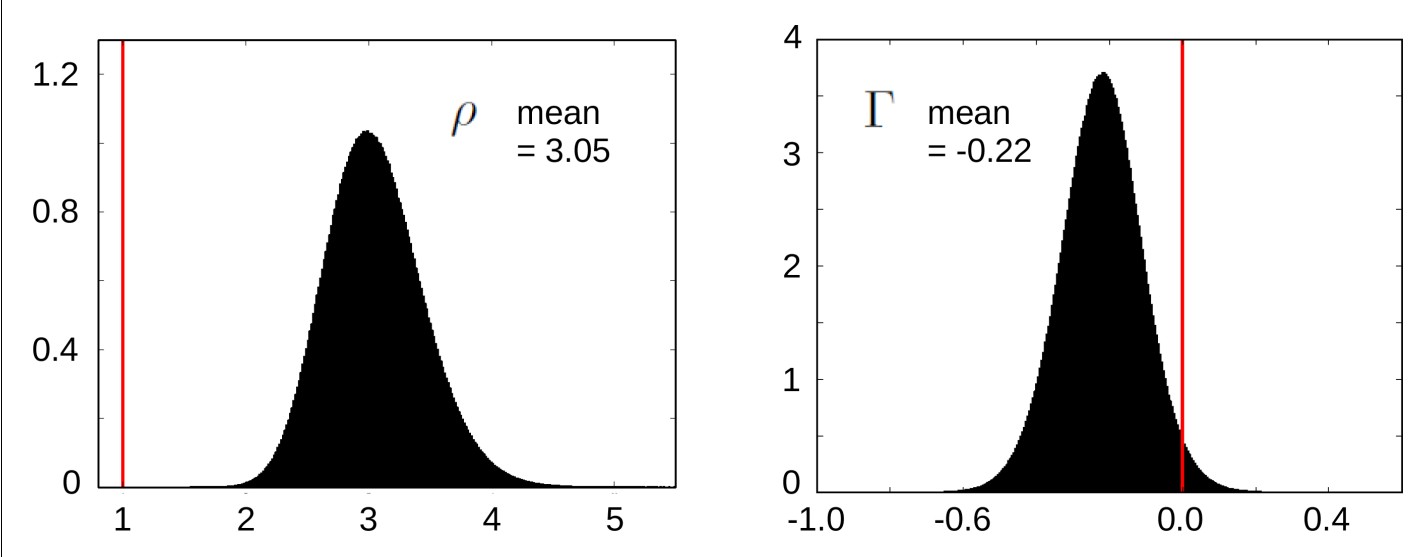

**Scheme 1.** Empirical distributions for $\rho$ (left) and $\Gamma$ (right).
DOI: https://doi.org/10.7554/eLife.41017.007

$\ell(i \to j) = a$. If we want to be clear about which graph is being referred to, we will place the graph symbol in a subscript, as in $i \to_G j$. If the system described by a graph can reach thermodynamic equilibrium, then the graph is reversible, so that if there is an edge $i \xrightarrow{a} j$ then there is also an edge $j \xrightarrow{b} i$. All the graphs considered in this paper are reversible.

## Path-independence and $R = 1$

As discussed in the main text, thermodynamic equilibrium can be characterised in several equivalent ways, including the existence of a free-energy landscape. For a reversible linear framework graph, this means that there is a function on the microstates, $\Phi(\nu(G)) \to \mathbf{R}$, which specifies the free-energy minima in *Figure 1C*. For any pair of reversible edges, $i \rightleftharpoons j$, this free-energy function satisfies,

$$\frac{\ell(i \to j)}{\ell(j \to i)} = \exp\left(\frac{\Phi(i) - \Phi(j)}{k_B T}\right). \tag{22}$$

*Equation 22* is the more general form of *Equation 3*. It tells us that the ratio of the edge labels depends only on the free-energy minima and is, therefore, a thermodynamic quantity.

Consider any path of reversible edges from microstate $i_1$ to microstate $i_K$, $i_1 \rightleftharpoons i_2 \rightleftharpoons \cdots \rightleftharpoons i_K$. The product of the label ratios along this path has a simple form because the terms inside the exponentials add together and thereby cancel out,

$$\frac{\ell(i_1 \to i_2)}{\ell(i_2 \to i_1)} \times \cdots \times \frac{\ell(i_{K-1} \to i_K)}{\ell(i_K \to i_{K-1})} = \exp\left(\frac{\Phi(i_1) - \Phi(i_2)}{k_B T}\right) \times \cdots \times \exp\left(\frac{\Phi(i_{K-1}) - \Phi(i_K)}{k_B T}\right)$$
$$= \exp\left(\frac{\Phi(i_1) - \Phi(i_2) + \cdots + \Phi(i_{K-1}) - \Phi(i_K)}{k_B T}\right) = \exp\left(\frac{\Phi(i_1) - \Phi(i_K)}{k_B T}\right). \tag{23}$$

We see that the product depends only on the initial microstate, $i_1$, and on the final microstate $i_K$. Accordingly, it does not depend on the path that is chosen between them.

The interpretation given to the quantities $K_i$ by Chen et al. is that they are the label ratios for the corresponding reversible edges in *Figure 1B* (*Chen et al., 2014*, Page S7), as described in *Equation 4*. It follows that the quantity $R$ in *Equation 1*, is the product of the label ratios for the upper path from **e** to **so** through **s**, which is $K_1 K_3$, divided by the product of the label ratios for the lower path from **e** to **so** through **o**, which is $K_2 K_4$. If the system is at thermodynamic equilibrium, then *Equation 23* shows that these products are identical, so that $K_1 K_3 = K_2 K_4$. Hence, at thermodynamic equilibrium, $R = 1$.

## The cycle condition and *Equation 6*

*Equation 23* has a particularly simple form if the path forms a cycle, so that $i_K = i_1$. In this case, $\Phi(i_1) = \Phi(i_K)$, so it follows from *Equation 23* by reorganizing the denominator on the left-hand side that,

$$\frac{\ell(i_1 \to i_2) \times \cdots \times \ell(i_{K-1} \to i_K)}{\ell(i_K \to i_{K-1}) \times \cdots \times \ell(i_2 \to i_1)} = 1 \,. \tag{24}$$

We see that the product of the edge labels going one way around the cycle equals the product of the edge labels going the other way around. A graph is at thermodynamic equilibrium if, and only if, *Equation 24* holds for any cycle of reversible edges in the graph. This is the equilibrium cycle condition.

For the graphs considered in this paper, whose edges come from binding and unbinding, a further simplification arises because the concentration terms cancel out above and below in *Equation 24*. This happens because any binding event that takes place along a cycle of reversible edges and incurs a TF concentration in the numerator of *Equation 24*, must eventually be undone by a corresponding unbinding event which incurs the same TF concentration in the denominator of *Equation 24*. Otherwise, the source microstate would acquire a new bound TF by going around the cycle, which is impossible. Hence, all concentration terms cancel out in *Equation 24*. Applying *Equation 24* to the unique cycle in *Figure 1B* then gives the quantity on the right-hand side of *Equation 6*. We see that the quantity $R$ in *Equation 1* calculates the cycle condition for the model in *Figure 1B*. Using *Equation 24*, we see once again that $R = 1$ at thermodynamic equilibrium.

## Binding-order asymmetry and *Equation 5*

As explained in the main text, the quantity $R$ in *Equation 1* does not measure the asymmetry in binding order of the two TFs. This binding-order asymmetry can be measured as follows.

Suppose that a system is represented by a graph and that the system is in microstate $i$ from which there is an outgoing edge $i \to j$. The probability that the system will take this edge depends on what other outgoing edges are available to it. These other edges have the form $i \to k$ where $k \neq j$. Each of these edges out of microstate $i$ have rates given by their corresponding labels, so the probability of taking $i \to j$ is just,

$$\frac{\ell(i \to j)}{\sum_{k \neq j} \ell(i \to k) + \ell(i \to j)} \,. \tag{25}$$

This quantity is the conditional probability of taking the edge $i \to j$, given that the system is in microstate $i$.

The conditional probability in *Equation 25* does not depend on time but the probability of being in microstate $i$ can change over time. This time-dependency can be avoided by assuming that the system is in steady state. It is then straightforward to calculate the probability at steady state of taking the lower path in *Figure 1B* from **e** to **o** to **so** :

$$\mathrm{Pr}(\mathbf{e}) \times \frac{k_2^+[O]}{k_1^+[S] + k_2^+[O]} \times \frac{k_4^+[S]}{k_4^+[S] + k_2^-} \,.$$

The first term gives the steady-state probability of being in microstate **e** , the second term gives the conditional probability from *Equation 25* of taking the edge $\mathbf{e} \to \mathbf{o}$, which puts the system in microstate **o** , from which the third term gives the conditional probability from *Equation 25* of taking the edge $\mathbf{o} \to \mathbf{so}$. Each event is independent of the others, so that the overall probability can be calculated as a product of the individual probabilities. Doing the same calculation for taking the upper path in *Figure 1B* from **e** to **s** to **so** :

$$\mathrm{Pr}(\mathbf{e}) \frac{k_1^+[S]}{k_1^+[S] + k_2^+[O]} \times \frac{k_3^+[O]}{k_3^+[O] + k_1^-} \,.$$

Taking the ratio of these two path probabilities gives *Equation 5*. Since we did not assume that the steady state was one of thermodynamic equilibrium, *Equation 5* gives a measure of steady-state binding-order asymmetry whether or not the graph is at thermodynamic equilibrium.

## Cooperativity

The term 'cooperativity' has several overlapping meanings in biology. For our purposes, it is a measure, at thermodynamic equilibrium, of how the binding of one TF is influenced by the binding of another TF. We know from *Equation 3*, or *Equation 22* above, that at thermodynamic equilibrium, the appropriate measure of binding of a TF $X$ is the ratio of the binding label, $k^+[X]$, to the unbinding label, $k^-$, in a DNA microstate graph. We can measure cooperativity by comparing this label ratio in different contexts. Consider the graph in *Figure 1B* for the binding of the TFs Sox2 and Oct4. The influence of Oct4 on the binding of Sox2 can be measured by comparing the label ratio for Sox2 binding when Oct4 is present, $k_4^+[S]/k_4^-$, to the label ratio for Sox2 binding when Oct4 is not present, $k_1^+[S]/k_1^-$. Let us call this quantity $\omega_{S,O}$,

$$\omega_{S,O} = \frac{k_4^+[S]}{k_4^-} \bigg/ \frac{k_1^+[S]}{k_1^-} = \frac{k_4^+ k_1^-}{k_4^- k_1^+} \,.$$

Note that this cooperativity is concentration independent. We can measure the influence of Sox2 on Oct4 binding in the same way and calculate $\omega_{O,S}$,

$$\omega_{O,S} = \frac{k_3^+[O]}{k_3^-} \bigg/ \frac{k_2^+[O]}{k_2^-} = \frac{k_3^+ k_2^-}{k_3^- k_2^+} \,.$$

But, at thermodynamic equilibrium, the equilibrium cycle condition (*Equation 6*) tells us that

$$\omega_{S,O}\omega_{O,S}^{-1} = \left(\frac{k_4^+ k_1^-}{k_4^- k_1^+}\right)\left(\frac{k_3^- k_2^+}{k_3^+ k_2^-}\right) = 1 \,.$$

Hence, $\omega_{S,O} = \omega_{O,S}$, so that the cooperative influence of Oct4 on Sox2 is the same as the cooperative influence of Oct4 on Sox2. This common quantity, which we call $\omega = \omega_{S,O} = \omega_{O,S}$, is the cooperativity between Sox2 and Oct4. The symmetry in the influence of the TFs on each other is a consequence of detailed balance at thermodynamic equilibrium. In fact, for the graph in *Figure 1B*, it is not hard to see that the condition

$$\omega_{S,O} = \omega_{O,S} \tag{26}$$

implies the equilibrium cycle condition, so that detailed balance is satisfied and the graph is at thermodynamic equilibrium. We will make use of this below.

If $\omega > 1$, the presence of either TF helps the other to bind; if $\omega < 1$, the presence of either TF hinders the other from binding; if $\omega = 1$, the presence of either TF makes no difference to how the other binds. The first condition is referred to, by a slight abuse of notation, as 'positive' cooperativity; the positivity resides not in the label ratios but in the free energy differences, $\Delta\Phi$ (*Figure 1D*). Similarly, the second condition is referred to as 'negative' cooperativity. The third condition is referred to as 'independence'.

The graphs in *Figure 2B and C* are described at thermodynamic equilibrium using label ratios and the cooperativity. When Sox2 is measured and Oct4 is induced (*Figure 2B*),

$$K_S = \frac{k_1^+[S]}{k_1^-} \ \ \text{and} \ \ K_O^i = \frac{k_2^+[O]}{k_2^-} \,,$$

with the appropriate concentrations, $[S]$ and $[O]$, for that context. When Oct4 is measured and Sox2 is induced (*Figure 2C*),

$$K_S^i = \frac{k_1^+[S]}{k_1^-} \ \ \text{and} \ \ K_O = \frac{k_2^+[O]}{k_2^-} \,,$$

with the appropriate concentrations, $[S]$ and $[O]$, for that context. In both contexts, the concentration-independent cooperativity is given by,

$$\omega = \frac{k_4^+ k_1^-}{k_4^- k_1^+} = \frac{k_3^+ k_2^-}{k_3^- k_2^+} \,.$$

## The single-locus model

### Calculating $\kappa_i$ and *Equations 15 and 16*

We consider the scenario in which Sox2 is measured and Oct4 is maximally induced (*Figure 2B*). $\kappa$ in *Equation 11* then corresponds to $\kappa = \kappa_4$. As explained in the main text, we need to calculate the average fraction of sites that are bound by Sox2 at steady-state without making the assumption, as Chen et al. did, that the Oct4 binding sites are saturated. With the model in *Figure 2B*, there is only one Sox2 binding site in any microstate and the average fraction of bound sites is the same as the average number of bound sites. The latter can be calculated by averaging the number of sites bound by Sox2 over the steady-state probabilities of the microstates. These probabilities can be calculated in turn using the prescription in *Equation 8*. As noted previously, this prescription gives the same result as equilibrium statistical mechanics. Taking a path from **e** to the microstate in question, *Equation 8* shows that,

$$\Pr(\mathbf{s}) = K_S \Pr(\mathbf{e})\,,\ \ \Pr(\mathbf{o}) = K_O^i \Pr(\mathbf{e})\,,\ \ \Pr(\mathbf{so}) = \omega K_S K_O^i \Pr(\mathbf{e})\,.$$

Since $\Pr(\mathbf{e}) + \Pr(\mathbf{s}) + \Pr(\mathbf{o}) + \Pr(\mathbf{so}) = 1$, we see that,

$$\Pr(\mathbf{e}) = \frac{1}{1 + K_O^i + K_S(1 + \omega K_O^i)}\,.$$

The term in the denominator is the partition function of equilibrium statistical mechanics. The average number of sites bound by Sox2 is then given by,

$$1 \cdot \Pr(\mathbf{s}) + 1 \cdot \Pr(\mathbf{so}) + 0 \cdot \Pr(\mathbf{e}) + 0 \cdot \Pr(\mathbf{o})\,,$$

from which we see that,

$$\frac{[DS]}{[D_S]_{tot}} = \frac{K_S(1 + \omega K_O^i)}{1 + K_O^i + K_S(1 + \omega K_O^i)}\,.$$

Using *Equation 12*, with $A = 1 + K_O^i$ and $B = K_S(1 + \omega K_O^i)$, and *Equation 13* we then find that

$$\kappa_4 = \frac{K_S(1 + \omega K_O^i)\alpha_S}{1 + K_O^i + K_S(1 + \omega K_O^i)(1 - \alpha_S)}\,, \tag{27}$$

as given in the main text.

To determine $\kappa_3$, we consider the scenario in which Oct4 is measured and Sox2 is maximally induced (*Figure 2C*) and calculate in a similar way the average number of sites bound by Oct4. We find that

$$\frac{[DO]}{[D_O]_{tot}} = \frac{K_O(1 + \omega K_S^i)}{1 + K_S^i + K_O(1 + \omega K_S^i)}\,.$$

Using *Equations 12 and 13* again, we see that

$$\kappa_3 = \frac{K_O(1 + \omega K_S^i)\alpha_O}{1 + K_S^i + K_O(1 + \omega K_S^i)(1 - \alpha_O)}\,. \tag{28}$$

The calculation of $\kappa_1$ was described in the main text and the calculation of $\kappa_2$ follows in a similar way. *Equations 15 and 16* are derived by using the resulting formulas.

## Steady-state probabilities away from thermodynamic equilibrium

If the single-locus model is away from thermodynamic equilibrium, then the prescription in *Equation 8* is no longer valid for calculating steady-state probabilities. However, non-equilibrium steady states can still be calculated analytically in the linear framework, as we briefly explain here. The procedure has been fully described, (*Ahsendorf et al., 2014*; *Estrada et al., 2016*; *Wong et al., 2018*), and these references should be consulted for more explanation and further details.

Away from equilibrium, the parameters in *Figure 2B and C* are no longer appropriate and must be replaced with the underlying edge labels, as in *Figure 1B*. For such a linear framework graph, $G$, which need not be reversible but should at least be strongly connected, the first step is to calculate

the quantities $\zeta_j^G$, where $j$ runs through each vertex in the graph. These quantities are given by a formula which derives from the Matrix Tree Theorem in graph theory,

$$\zeta_j^G = \sum_{T \in \Theta_j(G)} \left( \prod_{k \xrightarrow{a} l \in T} a \right). \tag{29}$$

In **Equation 29**, $\Theta_j(G)$ denotes the set of spanning trees of $G$ which are rooted at vertex $j$. A spanning tree, $T \in \Theta_j(G)$, is a subgraph of $G$ which includes every vertex in $G$ (spanning), has no cycles when edge directions are ignored (tree) and is rooted at $j$ if $j$ is the only vertex with no outgoing edges. **Equation 29** states that, to calculate $\zeta_j^G$, the labels on a spanning tree rooted at $j$ are multiplied together and these quantities are added together over all the spanning trees rooted at $j$.

The spanning trees for the same directed graph as in **Figure 1B** were previously enumerated in (**Estrada et al., 2016**, page 10). Although the labels are different, it is not hard to see that, following the prescription in **Equation 29**,

$$
\begin{aligned}
\zeta_\mathbf{e} &= k_2^- k_4^- k_3^+ [O] + k_1^- k_2^- k_4^- + k_1^- k_2^- k_3^- + k_1^- k_3^- k_4^+ [S] \\
\zeta_\mathbf{s} &= k_2^- k_4^- k_1^+ [S] + k_2^- k_3^- k_1^+ [S] + k_3^- k_1^+ k_4^+ [S]^2 + k_3^- k_2^+ k_4^+ [S][O] \\
\zeta_\mathbf{o} &= k_1^- k_3^- k_2^+ [O] + k_1^- k_4^- k_2^+ [O] + k_4^- k_2^+ k_3^+ [O]^2 + k_4^- k_1^+ k_3^+ [S][O] \\
\zeta_\mathbf{so} &= k_2^- k_1^+ k_3^+ [S][O] + k_1^+ k_3^+ k_4^+ [S]^2 [O] + k_2^+ k_3^+ k_4^+ [S][O]^2 + k_1^- k_2^+ k_4^+ [S][O]
\end{aligned}
$$

The $\zeta_j^G$ play a similar role away from thermodynamic equilibrium as the products of label ratios in **Equation 8** at equilibrium. The $\zeta_j^G$ are also proportional to the steady-state probabilities: $\mathrm{Pr}_G(j) = \lambda \zeta_j^G$, for some scalar factor $\lambda$. Accordingly, for a vertex $j$ in a general graph, $G$,

$$\mathrm{Pr}_G(j) = \frac{\zeta_j^G}{\sum_{i \in \nu(G)} \zeta_i^G}. \tag{30}$$

The denominator in **Equation 30** plays the role of a non-equilibrium partition function. This specification reduces to that of **Equation 8** if the graph is reversible and satisfies detailed balance (**Wong et al., 2018**). Using **Equations 29 and 30**, the steady-state probabilities away from thermodynamic equilibrium can be calculated in terms of the edge labels in the graph $G$ and this was used to derive the reciprocity plots in **Figure 2D and E** and **Figure 3B, C and D** (below).

## The genomic diversity model
### Average fraction of bound sites
The graph for the genomic diversity model in **Figure 3A** can be complicated, depending on the numbers of loci of types 1 and 2. However, provided the individual loci are independent of each other, the average fraction of $X$-specific sites bound by the TF $X$ can be easily calculated as follows. Let $G$ be a graph describing TF binding and let $\eta_X : \nu(G) \to \mathbf{R}$ be the function on microstates that counts the number of $X$-specific sites bound by $X$. For example, for the graphs in **Figure 2B** or 2C, $\eta_S(\mathbf{e}) = \eta_S(\mathbf{o}) = 0$ and $\eta_S(\mathbf{s}) = \eta_S(\mathbf{so}) = 1$. Let $\langle \eta_X \rangle_G$ denote the average of $\eta_X$ over the steady-state probabilities of the microstates in $G$, so that,

$$\langle \eta_X \rangle_G = \sum_{i \in \nu(G)} \eta_X(i) \mathrm{Pr}_G(i). \tag{31}$$

For the single-locus model, there is at most one molecule of each TF bound in the locus, so the average number of sites bound by that TF is the same as the average fraction bound. Hence, for the calculation of $\kappa_4$ for the single-locus model, we can write,

$$\frac{[DS]}{[D_S]_{tot}} = \langle \eta_S \rangle_G,$$

where $G$ is the graph in **Figure 2B**.

Now suppose that $G_1$ is the graph for one locus and $G_2$ is the graph for a second locus. Since we will be dealing throughout with averages over the steady-state probability distribution, the argument

which follows works irrespective of whether or not these graphs satisfy detailed balance and are at thermodynamic equilibrium. If the two loci are independent of each other, the overall graph for both loci considered together is the product graph $G_1 \times G_2$ introduced in *Ahsendorf et al. (2014)*. The vertices of $G_1 \times G_2$ are ordered pairs of vertices, $(i,j)$, one from each constituent graph, so that $i \in \nu(G_1)$ and $j \in \nu(G_2)$. If $i_1 \xrightarrow{a} i_2$ is a labeled edge in $G_1$, then, for any vertex $j \in \nu(G_2)$, $(i_1,j) \xrightarrow{a} (i_2,j)$ is a labeled edge in $G_1 \times G_2$. Similarly, if $j_1 \xrightarrow{b} j_2$ is a labelled edge in $G_2$, then, for any vertex $i \in \nu(G_1)$, $(i,j_1) \xrightarrow{b} (i,j_2)$ is a labeled edge in $G_1 \times G_2$. These constitute all the edges in $G_1 \times G_2$. This prescription for the edges captures the notion of independence: what happens in either constituent graph does not depend on the state of the other constituent graph. With this definition of the product graph, it can be shown that (*Ahsendorf et al., 2014*),

$$\mathrm{Pr}_{G_1 \times G_2}(i,j) = \mathrm{Pr}_{G_1}(i)\mathrm{Pr}_{G_2}(j),$$ 

(32)

as would be expected for probabilities under independence. Furthermore, it is evident that the number of sites bound by TF $X$ satisfies the addition formula,

$$\eta_X(i,j) = \eta_X(i) + \eta_X(j).$$

(33)

With these preliminaries, let us calculate the average number of sites bound by $X$ in $G_1 \times G_2$. Writing out the average according to *Equation 31*,

$$\langle \eta_X \rangle_{G_1 \times G_2} = \sum_{i \in \nu(G_1), j \in \nu(G_2)} \eta_X(i,j)\mathrm{Pr}_{G_1 \times G_2}(i,j).$$

Using *Equations 32 and 33*, the right-hand side can be rewritten as

$$\sum_{i \in \nu(G_1), j \in \nu(G_2)} (\eta_X(i) + \eta_X(j))\mathrm{Pr}_{G_1}(i)\mathrm{Pr}_{G_2}(j) = \sum_{i \in \nu(G_1), j \in \nu(G_2)} \eta_X(i)\mathrm{Pr}_{G_1}(i)\mathrm{Pr}_{G_2}(j) + \sum_{i \in \nu(G_1), j \in \nu(G_2)} \eta_X(j)\mathrm{Pr}_{G_1}(i)\mathrm{Pr}_{G_2}(j).$$

The first term on the right-hand side can be reorganized as

$$\sum_{j \in \nu(G_2)} \left( \sum_{i \in \nu(G_1)} \eta_X(i)\mathrm{Pr}_{G_1}(i) \right) \mathrm{Pr}_{G_2}(j) = \sum_{j \in \nu(G_2)} \langle \eta_X \rangle_{G_1} \mathrm{Pr}_{G_2}(j) = \langle \eta_X \rangle_{G_1},$$

where we have used *Equation. 31* again. A similar reduction can be made for the second term on the right-hand side above. Putting everything together yields the following observation, which takes into account that the only property of $\eta_X$ used in this argument was *Equation 33*.

## Lemma

If $G_1$ and $G_2$ are independent graphs, which do not have to satisfy detailed balance, and $f : \nu(G) \to \mathbf{R}$ is any function on graph microstates that is additive on the product graph, so that $f(i,j) = f(i) + f(j)$, then the average of $f$ is also additive,

$$\langle f \rangle_{G_1 \times G_2} = \langle f \rangle_{G_1} + \langle f \rangle_{G_2}.$$

(34)

Suppose now that across the genome we have $n_1$ loci of type 1, each described by the graph $G_1$ (*Figure 3A*, left), and $n_2$ loci of type 2, each described by the graph $G_2$ (*Figure 3A*, right). Assuming that the loci are independent of each other, we can form the overall product graph,

$$G = \underbrace{G_1 \times \cdots \times G_1}_{n_1 \text{ times}} \times \underbrace{G_2 \times \cdots \times G_2}_{n_2 \text{ times}}.$$

Applying *Equation 34* repeatedly with $f$ being $\eta_X$, we see that

$$\langle \eta_X \rangle_G = n_1 \langle \eta_X \rangle_{G_1} + n_2 \langle \eta_X \rangle_{G_2}.$$

Letting $l$ denote the proportion of type one loci, so that $l = n_1/(n_1 + n_2)$, we can divide the average number by $n_1 + n_2$ to get the average fraction, so that

$$\underbrace{\text{fraction}\,D_X\,\text{bound}}_{\text{genome}} = \frac{\langle\eta_X\rangle_G}{n_1+n_2} = l\langle\eta_X\rangle_{G_1} + (1-l)\langle\eta_X\rangle_{G_2}, \tag{35}$$

which is the formula used in the main text.

## Calculating $P_i$ and the reciprocity

As described in the main text, for each scenario indexed by $i = 1, \cdots, 4$, $P_i$ denotes the fraction of $X$-specific DNA binding sites that are bound by $X$ at steady state, or $[DX]/[D_X]_{tot}$. *Equation 35* can be used to calculate $P_i$. If we let $\mu = 1, 2$ be an index for the type of a locus, it will be helpful to let $P_{i,\mu}$ be the contribution to $P_i$ that comes from the locus of type $\mu$, so that $P_{i,\mu} = \langle\eta_X\rangle_{G_\mu}$, where $X = S$ for $i = 1, 4$ and $X = O$ for $i = 2, 3$. It follows from *Equation 35* that,

$$P_i = lP_{i,1} + (1-l)P_{i,2}. \tag{36}$$

The contributions $P_{i,\mu}$ can be calculated in the same way as for the single-locus model by using the corresponding graph $G_\mu$. For the locus of type $\mu$, we get, at thermodynamic equilibrium,

$$
\begin{aligned}
P_{1,\mu} &= \frac{K_{S,\mu}}{1+K_{S,\mu}} \\
P_{2,\mu} &= \frac{K_{O,\mu}}{1+K_{O,\mu}} \\
P_{3,\mu} &= \frac{K_{O,\mu}(1+\omega_\mu K_{S,\mu}^i)}{1+K_{S,\mu}^i+K_{O,\mu}(1+\omega_\mu K_{S,\mu}^i)} \\
P_{4,\mu} &= \frac{K_{S,\mu}(1+\omega_\mu K_{O,\mu}^i)}{1+K_{O,\mu}^i+K_{S,\mu}(1+\omega_\mu K_{O,\mu}^i)}
\end{aligned}
$$

It follows that,

$$
\begin{aligned}
\frac{P_{4,\mu}}{P_{1,\mu}} &= \frac{1+\omega_\mu K_{O,\mu}^i+K_{S,\mu}(1+\omega_\mu K_{O,\mu}^i)}{1+K_{O,\mu}^i+K_{S,\mu}(1+\omega_\mu K_{O,\mu}^i)} \\
\frac{P_{3,\mu}}{P_{2,\mu}} &= \frac{1+\omega_\mu K_{S,\mu}^i+(1+K_{O,\mu})\omega_\mu K_{S,\mu}^i}{1+K_{S,\mu}^i+K_{O,\mu}(1+\omega_\mu K_{S,\mu}^i)}
\end{aligned}
\tag{37}
$$

It is evident from *Equation 37* that if $\omega_\mu > 1$, then,

$$P_{4,\mu} > P_{1,\mu} \;\text{ and }\; P_{3,\mu} > P_{2,\mu},$$

while if $\omega_\mu < 1$, then,

$$P_{4,\mu} < P_{1,\mu} \;\text{ and }\; P_{3,\mu} < P_{2,\mu}.$$

Using *Equation 36* we see that, when both types of loci are at thermodynamic equilibrium and exhibit the same sign for their cooperativity,

$$\underbrace{\frac{P_4}{P_1}>1, \frac{P_3}{P_2}>1}_{\text{if }\omega_1>1,\,\omega_2>1} \;\text{ or }\; \underbrace{\frac{P_4}{P_1}<1, \frac{P_3}{P_2}<1}_{\text{if }\omega_1<1,\,\omega_2<1}.$$

It follows that the reciprocity, as defined in *Equation 20*, is always positive.

## Parameter values for the reciprocity plots

The reciprocity plots in *Figure 2D and E* and *Figure 3B, C and D* were calculated using the non-equilibrium steady-state probabilities, as specified above, so as to allow analysis of both the non-equilibrium and equilibrium regimes. For this, we reverted to the parameters for the graph in *Figure 1B*, which underlies both the single-locus and genomic-diversity models. As noted above, the non-equilibrium steady-state probabilities in *Equation 30* revert to the equilibrium ones determined by *Equation 8* if the graph satisfies the equilibrium cycle condition. We calculated the quantities $\omega_{S,O}$ and $\omega_{O,S}$, as listed in the tables below, and used $\omega_{S,O} = \omega_{O,S}$ (*Equation 26*) to confirm the equilibrium cycle condition. In such cases, the common value, $\omega_{S,O} = \omega_{O,S}$ is the cooperativity. The reciprocity in *Equation 20* involves all four experimental scenarios and therefore requires four concentration values for the measured and induced TFs in the two cell lines. The concentrations of the measured TFs

are denoted by $[S]$ and $[O]$ and given numerical values in the tables below. The concentrations of the induced TFs are denoted by $[Sox2^i]$ and $[Oct4^i]$ and treated as variables in the plots. The parameter values are in arbitrary units, subject to the on-rates, $k_i^+$, having dimensions of (concentration $\times$ time)$^{-1}$, the off-rates having dimensions of (time)$^{-1}$ and $[S]$ and $[O]$ having dimensions of (concentration). The reciprocity plots were created in a Mathematica notebook, available on request.

The parameter values for the single-locus model are as follows, with those for **Figure 2D** being at thermodynamic equilibrium.

| Figure 2D | | Figure 2E | |
| --- | --- | --- | --- |
| Parameters | Value | Parameters | Value |
| $k_1^+$ | 3 | $k_1^+$ | 3 |
| $k_2^+$ | 3 | $k_2^+$ | 3 |
| $k_3^+$ | 6 | $k_3^+$ | 6 |
| $k_4^+$ | 6 | $k_4^+$ | 6 |
| $k_1^-$ | 2 | $k_1^-$ | 2 |
| $k_2^-$ | 4 | $k_2^-$ | 4 |
| $k_3^-$ | 2 | $k_3^-$ | 2 |
| $k_4^-$ | 1 | $k_4^-$ | 12 |
| $[S]$ | 1 | $[S]$ | 1 |
| $[O]$ | 1 | $[O]$ | 1 |
| $\omega_{S,O}$ | 4 | $\omega_{S,O}$ | 1/3 |
| $\omega_{O,S}$ | 4 | $\omega_{O,S}$ | 4 |

The parameter values for the genomic-diversity model are as follows, with those for **Figure 3B and D** being at thermodynamic equilibrium. The rate constants for type 1 loci are indicated by (I) and for type 2 loci by (II), with the same convention for $\omega_{S,O}$ and $\omega_{O,S}$. $l$ is the proportion of type 1 loci.

| Figure 3B | | Figure 3C | | Figure 3D | |
| --- | --- | --- | --- | --- | --- |
| Parameters | Value | Parameters | Value | Parameters | Value |
| $k_1^+$ (I) | 1 | $k_1^+$ (I) | 1 | $k_1^+$ (I) | $10^{-0.2}$ |
| $k_2^+$ (I) | 1 | $k_2^+$ (I) | 1 | $k_2^+$ (I) | $10^{-0.3}$ |
| $k_3^+$ (I) | 5 | $k_3^+$ (I) | 5 | $k_3^+$ (I) | $10^{+0.7}$ |
| $k_4^+$ (I) | 5 | $k_4^+$ (I) | 5 | $k_4^+$ (I) | $10^{+0.8}$ |
| $k_1^-$ (I) | 8 | $k_1^-$ (I) | 8 | $k_1^-$ (I) | 1 |
| $k_2^-$ (I) | 8 | $k_2^-$ (I) | 8 | $k_2^-$ (I) | 1 |
| $k_3^-$ (I) | 3 | $k_3^-$ (I) | 3 | $k_3^-$ (I) | 1 |
| $k_4^-$ (I) | 3 | $k_4^-$ (I) | 3 | $k_4^-$ (I) | 1 |
| $k_1^+$ (II) | 10 | $k_1^+$ (II) | 10 | $k_1^+$ (II) | $10^{-0.9}$ |
| $k_2^+$ (II) | 10 | $k_2^+$ (II) | 10 | $k_2^+$ (II) | $10^{-0.9}$ |
| $k_3^+$ (II) | 11 | $k_3^+$ (II) | 11 | $k_3^+$ (II) | $10^{-1.9}$ |
| $k_4^+$ (II) | 11 | $k_4^+$ (II) | 11 | $k_4^+$ (II) | $10^{-1.9}$ |
| $k_1^-$ (II) | 1 | $k_1^-$ (II) | 1 | $k_1^-$ (II) | 1 |
| $k_2^-$ (II) | 1 | $k_2^-$ (II) | 1 | $k_2^-$ (II) | 1 |
| $k_3^-$ (II) | 1 | $k_3^-$ (II) | 1 | $k_3^-$ (II) | 1 |
| $k_4^-$ (II) | 1 | $k_4^-$ (II) | 40 | $k_4^-$ (II) | 1 |

*Continued on next page*

*Continued*

| Figure 3B | | Figure 3C | | Figure 3D | |
|---|---|---|---|---|---|
| Parameters | Value | Parameters | Value | Parameters | Value |
| $[S]$ | 1 | $[S]$ | 1 | $[S]$ | 6 |
| $[O]$ | 1 | $[O]$ | 1 | $[O]$ | 0.1 |
| $l$ | 0.67 | $l$ | 0.67 | $l$ | 0.3 |
| $\omega_{S,O}(\text{I})$ | 40/3 | $\omega_{S,O}(\text{I})$ | 40/3 | $\omega_{S,O}(\text{I})$ | 10 |
| $\omega_{O,S}(\text{I})$ | 40/3 | $\omega_{O,S}(\text{I})$ | 40/3 | $\omega_{O,S}(\text{I})$ | 10 |
| $\omega_{S,O}(\text{II})$ | 11/10 | $\omega_{S,O}(\text{II})$ | 11/400 | $\omega_{O,S}(\text{I})$ | 0.1 |
| $\omega_{O,S}(\text{II})$ | 11/10 | $\omega_{O,S}(\text{II})$ | 11/10 | $\omega_{O,S}(\text{II})$ | 0.1 |

## Acknowledgements

We thank Rosa Martinez-Corral for helpful comments on earlier drafts. We are also grateful to Hinrich Boeger and an anonymous reviewer for thoughtful suggestions which improved the clarity of the exposition. JWB and JG were supported by NSF 1462629.

## Additional information

### Funding

| Funder | Grant reference number | Author |
|---|---|---|
| National Science Foundation | 1462629 | John W Biddle<br>Jeremy Gunawardena |

The funders had no role in study design, data collection and interpretation, or the decision to submit the work for publication.

### Author contributions

John W Biddle, Formal analysis, Methodology, Writing—review and editing; Maximilian Nguyen, Conceptualization, Formal analysis, Writing—review and editing; Jeremy Gunawardena, Conceptualization, Formal analysis, Supervision, Methodology, Writing—original draft

### Author ORCIDs

Jeremy Gunawardena (iD) https://orcid.org/0000-0002-7280-1152

### Decision letter and Author response

Decision letter https://doi.org/10.7554/eLife.41017.009
Author response https://doi.org/10.7554/eLife.41017.010

## Additional files

### Supplementary files

• Transparent reporting form
DOI: https://doi.org/10.7554/eLife.41017.008

### Data availability

All data analysed during this study are included in the manuscript and supporting files. No new data were generated.

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
