## [Decision Letter]

Thank you for submitting your article "Negative reciprocity, not ordered assembly, underlies the interaction of Sox2 and Oct4 on DNA" for consideration by *eLife*. Your article has been reviewed by two peer reviewers, and the evaluation has been overseen by Naama Barkai as the Reviewing and Senior Editor.

The reviewers have discussed the reviews with one another and the Reviewing Editor has drafted this decision to help you prepare a revised submission. As you will see below, only minor revisions were requested: please address them.

*Reviewer #1:*

In this manuscript by Biddle et al., the authors undertake a re-analysis of published single-molecule imaging data of transcription factor mobility in the nucleus. The published data, from Chen et al., led to the conclusion that one could determine the order of assembly of transcription complexes. Specifically, their data suggested that Sox2 binding preceded Oct4 binding on DNA. The present paper was a challenging one to review. It is written in a very pedagogical, non-traditional manner and aims to correct what appear to be serious errors in the previous analysis. At the same time, the theory in this work goes further and sheds new insight on the biophysics of gene regulation. More specifically, using an approach that largely consists of solving rate equations, they arrive at this notion of reciprocity, which speaks to the interaction between factors such as Sox2 and Oct4. The paper is geared toward biologists, not physicists, and overall I found the message to be important and also conveyed in a lucid manner. However, I emphasize that the work is almost a hybrid between an essay, research paper, and review. The paper drifts between undergraduate level descriptions of statistical mechanics and direct criticism of recently published work. Yet, because of the subject material, the style of writing, and the biological significance of the finding, I think *eLife* could be a unique fit.

Major comments:

1) The math is straightforward, and after going through the entire logic of the analysis (there are 20 equations in the main text alone), I don't find anything objectionable. The key point is this notion of cooperativity ('*ω*') and the related 'reciprocity' parameter. I thought the authors did a good job of developing this concept.

2) Essentially, this reciprocity parameter replaces the order of assembly description provided by Chen. Biddle et al. derive a rigorous steady state analysis, which has a more restricted interpretation: when Oct4 expression increases, Sox2 binding goes down. But when Sox2 expression increases, Oct4 binding goes up. So there is an asymmetry, which led Chen and coworkers to conclude using incorrect analysis but roughly correct phenomenological interpretation that under these experimental conditions, Sox2 enables binding of Oct4. I agree with Biddle et al., that this phenomenological interpretation masks the underlying mechanistic details which allow for many different potential binding cooperativities, both positive and negative. What I wonder is: could this conclusion have been correctly reached without any model? In other words, just from looking at the raw data in cell lines that there was an asymmetric behavior depending on whether Oct4 or Sox2 was induced?

3) Perhaps the most interesting conclusion of the paper is that genomic diversity (i.e. different types of binding sites/cooperativities) can generate this apparent reciprocity even at thermodynamic equilibrium, complicating any conclusion about the extent of energy involvement in the assembly of transcriptional complexes.

4) The language in Chen et al. may have been imprecise, but I don't think there was ever an assumption of thermodynamic equilibrium.

5) At the heart of this question is the textbook description of ordered assembly of transcription complexes. Such mental constructs require ATP to provide directionality, even though many molecular biologists do not explicitly acknowledge this fact but rather tend to think and speak in terms of thermal equilibrium. For this reason, these pictures have been hard to reconcile with the short dwell times of TFs observed by microscopy. There is a view that it would be hard for short interactions to be 'functional.' However, this statement again assumes an equilibrium bias. With ATP hydrolysis anything is possible: TF could have very short dwell times because they are only removed after they have done their job. Many conceptual models have been proposed to account for this seeming paradox. Ultimately, based on this analysis the conclusions of Chen et al. are ambiguous, leaving the fundamental question of 'ordered assembly' up for debate. Overall, this paper strikes me as a good fit for *eLife*, but it may need an editorial touch.

*Reviewer #2:*

"Negative reciprocity, not ordered assembly, underlies the interaction of Sox2 and Oct4 on DNA" by Biddle and colleagues provides an important theoretical contribution to the biophysics of gene regulation. I strongly support its publication.

The point of departure for the authors' analysis – based on their linear framework approach (Gunawardena, 2012. Mirzaev and Gunawardena, 2013) – is a paper by Chen et al.: "Single Molecule Dynamics of Enhanceosome Assembly in Embryonic Stem Cells" (Cell 156, 1274-1285, 2014).

As a result of single molecule tracking of the transcription factors Sox2 and Oct4 in live cells by fluorescence microscopy, Chen and colleagues claimed that "Unexpectedly, we observed that Sox2 is the lead TF (transcription factor) that assists Oct4 to assemble on its in vivo targets in a hierarchically ordered mechanism."

Chen et al.'s claim was based not on observation, of course, but on theoretical interpretations of their data; interpretations that Biddle et al. show were mistaken: The factor "R", calculated by Chen et al. from their microscopic measurements, does not correspond to the ratio of steady state probabilities for two alternative assembly pathways of TF on the DNA, as Chen at al. incorrectly suggest. Their finding of R > 1 does therefore not indicate greater probability of one assembly pathway over the other. Rather, Biddle et al. show that R > 1 implies violation of the detailed balance condition for thermodynamic equilibrium, and thus indicates energy expenditure "behind the scenes". This would have been a finding of great interest had Chen at al. not confused theoretical and observational viewpoints: the "TF graph" and "DNA graph" viewpoint; Chen et al.'s R – obtained from measurements taken from the TF and not DNA viewpoint – is uninformative.

Biddle et al. however come to the rescue of Chen et al.'s analysis by deriving formulae that relate the quantities of both viewpoints and by introducing a novel concept: reciprocity, Γ, which is calculated from Chen et al.'s measurements in a straight forward manner. In equilibrium, Γ > 0, if both TFs bind to DNA cooperatively – either positively or negatively – or else Γ = 0.

However, Chen et al.'s data suggest Γ < 0. Biddle et al. go on to show that Γ < 0 either indicates that one TF (Sox2) positively affects binding of the other (Oct4), whereas the latter (Oct4) negatively affects binding of the first (Sox2) – remarkably, this asymmetry requires energy expenditure, for it is incompatible with thermodynamic equilibrium – or mixed positive and negative cooperativity of both TFs at different DNA binding sites.

The point of departure for Biddle et al. is their well-founded criticism of one aspect of the work of Chen et al. – other aspects of this work are worthy of critical analysis, too, but this is not the focus of their analysis. However, the scope of Biddle et al.'s analysis is not limited to this criticism. This paper makes an important theoretical contribution to the problem of transcriptional regulation. The text is clearly written; its arguments are transparent. This thoughtful analysis is illuminating in other regards, too: it painfully highlights the extent to which deeper theoretical thought is absent from the pages of our journals, and how easily, therefore, both reviewers and editors may succumb to authority rather than scientific argument.

Minor Comments:

I only have one criticism, or rather request. Despite all care given to the matter, I found myself nonetheless confused at times by the reinterpretation of Chen et al.'s Ki's. The confusion, in part, stems from using the same symbol K*i for different entities in equations (4) and (9), which propagates Chen et al.'s confusion. It might be better, I think, to use different symbols in both instances. Indeed the authors do so later in their text, by introducing KS, KO, etc., which however are not defined in the main text but in the Materials and Methods section alone.

In the same vein, "at face value" ("treating at face value Chen et al.'s calculations of R*") may better be specified, e.g. by pointing out that Chen et al. confused K*1 with KS etc. - by (unconsciously?) switching from the TF viewpoint to the DNA viewpoint.

Similarly, confusion arises in the Introduction, where it is stated first that R ≠ 0 implies energy expenditure away from equilibrium, for it violates the cycle condition, i.e., detailed balance, of thermodynamic equilibrium, and later that R ≠ 0 is compatible with equilibrium; that R in both instances is not the same is easily be lost upon first reading.

---

## [Author Response]

Reviewer #1:

[…] Yet, because of the subject material, the style of writing, and the biological significance of the finding, I think eLife could be a unique fit.

We are attempting several things in this paper: correcting errors in Chen et al; re-analysing their data and deriving new biological insights from them; and attempting to educate the field about the biophysics of gene regulation. This may account for the “hybrid” quality that the reviewer describes. We feel that this style is justified because the errors in Chen et al. have remained unnoticed in the literature for over four years despite the paper being widely cited. This suggests to us that an undergraduate-level description of statistical mechanics is necessary for the conclusions to be understandable and compelling to a biological audience. We fully agree with the reviewer that *eLife* offers a unique fit for this kind of paper.

What I wonder is: could this conclusion have been correctly reached without any model? In other words, just from looking at the raw data in cell lines that there was an asymmetric behavior depending on whether Oct4 or Sox2 was induced?

The reviewer is correct that the raw data in cell lines shows asymmetric behaviour depending on whether Oct4 or *Sox2* is induced (as in Figure 5 of Chen et al.). However, the raw data comes from observing the TFs. We do not see how conclusions can be drawn about what happens on DNA without some model of what is happening on DNA and some rigorous way of moving from the TF to the DNA viewpoint. This was a key point in our analysis. Hence, we do not believe our conclusions about reciprocity could have been achieved without a model.

Perhaps the most interesting conclusion of the paper is that genomic diversity (i.e. different types of binding sites/cooperativities) can generate this apparent reciprocity even at thermodynamic equilibrium, complicating any conclusion about the extent of energy involvement in the assembly of transcriptional complexes.

Yes! This was a surprise to us as well.

The language in Chen et al. may have been imprecise, but I don't think there was ever an assumption of thermodynamic equilibrium.

Chen et al's calculation of the quantity that we have called R in Equation 1 is given on page S6 of their Supplementary Information. It refers to a diagram that is effectively the same as our Figure 1B and is followed by the remark, “When the above reactions are at equilibrium... “

Such mental constructs require ATP to provide directionality, even though many molecular biologists do not explicitly acknowledge this fact but rather tend to think and speak in terms of thermal equilibrium. For this reason, these pictures have been hard to reconcile with the short dwell times of TFs observed by microscopy. There is a view that it would be hard for short interactions to be 'functional.' However, this statement again assumes an equilibrium bias. With ATP hydrolysis anything is possible…

We strongly agree with the reviewer and hope that our paper will encourage more awareness of the role played by energy expenditure in gene regulation.

Reviewer #2:

We are most grateful to the reviewer for the strong support.

I only have one criticism, or rather request. Despite all care given to the matter, I found myself nonetheless confused at times by the reinterpretation of Chen et al.'s Ki's. The confusion, in part, stems from using the same symbol K*i for different entities in equations (4) and (9), which propagates Chen et al.'s confusion. It might be better, I think, to use different symbols in both instances.

This is a very valid point and we apologise for our poor judgement about it. We had introduced an asterisk notation, as in R*, to denote the convention that we followed, of taking the ratio of binding rates to unbinding rates, in contrast to the opposite convention followed by Chen et al. We felt that further notation to distinguish between the TF and DNA viewpoints would diminish the paper's readability. However, the reviewer is absolutely correct that the second issue is fundamental, while the first issue is merely cosmetic. Accordingly, we have reversed our priorities. We have delegated the convention on rate ratios to a note at the start of the Results, to which we have referred in the Introduction, and we have dropped the use of asterisks. We have introduced notation based on the Greek symbols \kappa and \rho for the quantities arising from the TF viewpoint (Equation 2), thereby clearly distinguishing them from the quantities K and R arising from the DNA viewpoint. We are grateful to the reviewer for this very helpful criticism. We note that these changes had to be propagated throughout the paper and necessitated further changes of notation (for instance, we replaced \kappa by \mu).

Indeed the authors do so later in their text, by introducing KS, KO, etc., which however are not defined in the main text but in the Materials and Methods section alone.

Actually, the symbols K_S, K_O are defined in the main text.

In the same vein, "at face value" ("treating at face value Chen et al.'s calculations of R*") may better be specified, e.g. by pointing out that Chen et al. confused K*1 with KS etc.- by (unconsciously?) switching from the TF viewpoint to the DNA viewpoint.

An earlier response should have addressed this confusion. We have altered the text in question to reflect this by saying that “we can be fooled into thinking that the model in Figure 1A is away from thermodynamic equilibrium by Chen et al’s conflation of R with \rho”.

Similarly, confusion arises in the Introduction, where it is stated first that R ≠ 0 implies energy expenditure away from equilibrium, for it violates the cycle condition, i.e., detailed balance, of thermodynamic equilibrium, and later that R ≠ 0 is compatible with equilibrium; that R in both instances is not the same is easily be lost upon first reading.

Again, the changes made should have dealt with this further confusion. The second mention of compatibility with thermodynamic equilibrium is in terms of \rho, not R.